# DomED: Redesigning Ensemble Distillation for Domain Generalization

## Abstract

Domain generalization aims to improve model performance on unseen, out-of-distribution (OOD) domains, yet existing methods often overlook the crucial aspect of uncertainty quantification in their predictions. While ensemble learning combined with knowledge distillation offers a promising avenue for enhancing both model accuracy and uncertainty estimation without incurring significant computational overhead at inference time, this approach remains largely unexplored in the context of domain generalization. In this work, we systematically investigate different ensemble and distillation strategies for domain generalization tasks and design a tailored data allocation scheme to enhance OOD generalization as well as reduce computational cost. Our approach trains base models on distinct subsets of domains and performs distillation on complementary subsets, thereby fostering model diversity and training efficiency. Furthermore, we develop a novel technique that decouples uncertainty distillation from the standard distillation process, enabling the accurate distillation of uncertainty estimation capabilities without compromising model accuracy. Our proposed method, *Domain-aware Ensemble Distillation* (DomED), is extensively evaluated against state-of-the-art domain generalization and ensemble distillation techniques across multiple benchmarks, achieving competitive accuracies and substantially improved uncertainty estimates.

## 1 Introduction

A fundamental assumption in many machine learning techniques is that training and test data are drawn from the same distribution. However, this assumption often fails in real-world scenarios, where models trained in one environment may be deployed in a different environment, leading to a distributional shift. Domain generalization (DG) (Shankar et al., 2018; Zhou et al., 2020) addresses this challenge by training a model on multiple source domains such that it can better generalize to unseen target domains.

Existing DG methods primarily focus on learning domain-invariant representations, employing techniques like explicit feature alignment (Ghifary et al., 2015; Maniyar et al., 2020), domain adversarial learning (Du et al., 2021; Li et al., 2018b), and feature disentanglement (Mahajan et al., 2021; Zhang et al., 2022). While effective, these methods still exhibit limited generalization performance on out-of-distribution (OOD) data, often not significantly outperforming carefully-tuned empirical risk minimization (ERM) (Gulrajani & Lopez-Paz, 2020). More critically, they tend to produce overconfident yet erroneous predictions on OOD data (Ovadia et al., 2019), rendering their predictions unreliable. This highlights the need to consider both prediction accuracy and uncertainty estimation in developing robust DG methods.

Ensemble methods, which average the outputs of multiple models, are known to improve generalization and provide more accurate uncertainty estimates (Opitz & Maclin, 1999; Dietterich, 2000; Zhou et al., 2002; Rokach, 2010; Lakshminarayanan et al., 2017), particularly for epistemic uncertainty. However, the computational and memory overhead of using ensembles at inference time can be prohibitive. Knowledge distillation offers a solution by compressing an ensemble into a single, efficient model while preserving its uncertainty estimation capabilities (Tran et al., 2020; Malinin et al., 2019; Ferianc & Rodrigues, 2022). Recent work has explored self-distilling an ensemble of the output logits from training data with the same label (Lee et al., 2022), but the potential of combining explicit ensemble learning with uncertainty-aware distillation remains underexplored in the context of DG. Specifically, a principled framework for leveraging domain labels to construct diverse experts

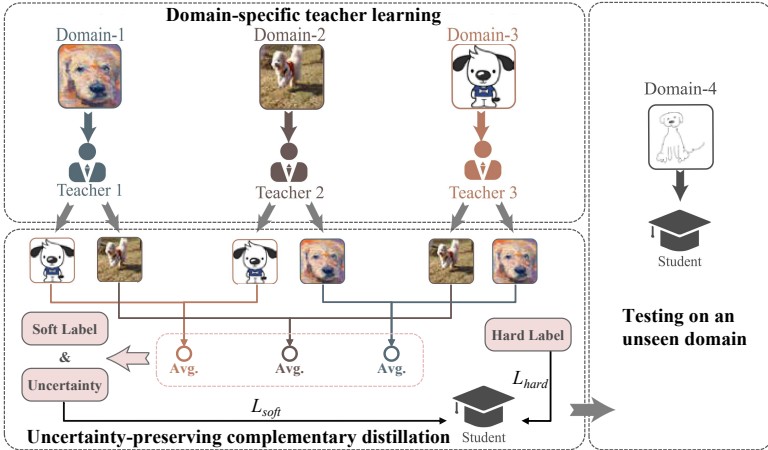

Figure 1: Illustration of DomED. Each teacher model is trained on a specific source domain. The teachers collectively make predictions on data complementary to their respective training data, such that their generalization ability along with their inherent uncertainty information can be distilled to a student model. After distillation, the student model is evaluated on a previously unseen target domain.

remains elusive, and simultaneously preserving prediction and uncertainty estimation capabilities presents a non-trivial optimization conflict (Ryabinin et al., 2021).

In this work, we aim to develop a tailored ensemble learning and knowledge distillation scheme for domain generalization. Although it is not fully understood how ensemble learning improves the test-time performance of deep neural networks, recent work suggests that training multiple models to exploit the "multi-view" structure in data is crucial for the success of ensemble methods (Allen-Zhu & Li, 2020). This aligns with the understanding that model diversity is key in ensemble learning (Brown et al., 2005; Nam et al., 2021a). Unlike regular classification tasks, DG provides domain labels for each data sample, presenting a natural opportunity for multi-view learning. We propose to train the base models of an ensemble on different, non-overlapping subsets of domains. This enhances model diversity while reducing the training cost of individual models. We then evaluate and compare different data allocation schemes for ensembling and distillation to identify the optimal scheme. To the best of our knowledge, we are the first to systematically investigate the possible data allocation schemes for adapting ensemble distillation to the particular setting of domain generalization.

Beyond achieving high accuracy on unseen domains, our goal is to distill the uncertainty estimation capability of ensembles. A common approach is to train a prior network to output a conjugate prior (e.g., a Dirichlet distribution for classification tasks) that captures the output distributions of the base models (Malinin & Gales, 2018; Malinin et al., 2019). However, we find that this approach significantly degrades model accuracy in DG compared to standard distillation (Hinton et al., 2015). To address this, we introduce a novel technique that decouples uncertainty distillation from standard distillation, allowing for accurate model predictions and uncertainty estimates simultaneously. We refer to our approach as *Domain-aware Ensemble Distillation* (DomED), as illustrated in Figure 1.

Our main contributions are summarized as follows:

- We explore tailored ensemble and distillation strategies for domain generalization tasks, and develop a novel data allocation scheme that trains and distills base models on complementary domains, which enhances model diversity and training efficiency.

- We identify that the commonly used uncertainty distillation method degrades the accuracy of an ensemble after distillation in the context of DG. We address this by proposing a decoupled distillation technique that preserves both mean prediction accuracy and model uncertainty.

- We conduct extensive experiments to compare different data allocation schemes and evaluate our approach, DomED, on multiple domain generalization benchmarks. Our results demonstrate that DomED achieves competitive accuracies and significantly improved uncertainty quantification compared to existing methods.

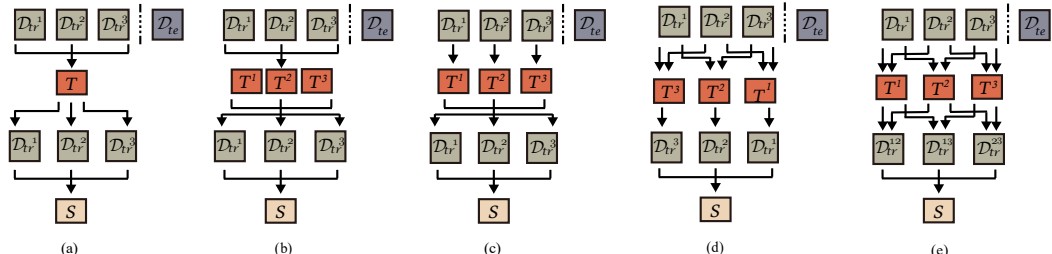

Figure 2: Comparison of different data allocation schemes for ensemble learning and knowledge distillation. Note that these are different from the DomED scheme presented in Figure 1. Given $M$ training domains, (a) self-distillation from a single teacher model; (b) standard ensemble distillation (all teachers trained and distilled on all domains); (c) includes $M$ single-domain teachers, distilled on all domains; (d) includes $M$ teachers, each trained on $M-1$ domains and distilled on the remaining domain; and (e) includes $M$ teachers, each trained and distilled on the same $M-1$ domains.

## 2 METHODS

We present our ensemble learning and knowledge distillation methods for domain generalization. First, we formally define the domain generalization problem and introduce the relevant notation. We then discuss various ensembling and distillation strategies and develop a tailored scheme for domain generalization that achieves high model accuracy and reduced training cost. Based on this scheme, we further propose a novel uncertainty-preserving distillation method, which decouples uncertainty distillation from the standard distillation process, to simultaneously achieve accurate model predictions and uncertainty estimates.

### 2.1 PROBLEM DEFINITION

We consider a domain generalization problem with $M$ source domains (training domains), whose union is denoted as $\mathcal{D}_{tr} = \bigcup_{m=1}^{M} \mathcal{D}_{tr}^m$, and one unseen target domain (test domain), $\mathcal{D}_{te}$. Each source domain, $\mathcal{D}_{tr}^m$, contains $N_m$ independent and identically distributed (i.i.d.) labeled training samples, i.e., $\mathcal{D}_{tr}^m := \{(x_i^m, y_i^m)\}_{i=1}^{N_m}$. Similarly, the target domain comprises $N_{te}$ unlabeled i.i.d. samples, $\mathcal{D}_{te} := \{x_j\}_{j=1}^{N_{te}}$. While all domains share a common feature space $\mathcal{X}$ and label space $\mathcal{Y}$, the core challenge is that the data distribution varies across domains. In image classification, for instance, domains might represent different visual styles (e.g., photos, sketches) but share the same set of object categories. The objective is to learn a function $f : \mathcal{X} \longrightarrow \mathcal{Y}$ using only data from the source domains that can effectively generalize to the unseen target domain.

### 2.2 ENSEMBLE AND DISTILLATION STRATEGIES

In a regular supervised learning problem that has i.i.d. data for both training and testing, the base models of an ensemble are usually trained on the same dataset, and model diversity is introduced only by the independent initialization of their parameters (Allen-Zhu & Li, 2020). Moreover, the distillation of the ensemble is also done on the same set of training data. In contrast, a domain generalization task splits data into multiple domains, making it possible to increase model diversity by training base models on different subsets of domains. Furthermore, for each base model, it is also possible to use a different set of data than its training data for distillation. As shown in Figure 1 and Figure 2, we consider six representative training data allocation schemes, and compare them empirically in Section 3. Among them, Figure 2(a) depicts the self-distillation scheme, and Figure 2(b) depicts the regular ensembling and distillation scheme. For fair comparison, we assume an ensemble of $M$ base models (one for each source domain), $\{T^m\}_{m=1}^{M}$. In this section, we discuss in detail the scheme employed by DomED (as shown in Figure 1) due to its superior empirical performance and relatively low computational cost. In the following, we also refer to the base models of an ensemble and the distilled model, respectively, as teacher models and student model.

**Domain-specific teacher models.** The data allocation scheme of DomED is designed to maximize teacher diversity by training each of the $M$ teacher models on a single, distinct source domain. This strategy treats each domain as a unique "view" of the data, encouraging the models to develop different specializations. Specifically, each teacher model, $T^m$, is a neural network parameterized by weights $\theta_T^m$ and is trained exclusively on its corresponding domain, $\mathcal{D}_{tr}^m$. The complete set of

teacher parameters is denoted as $\theta_T = \{\theta_T^m\}_{m=1}^M$. For classification tasks, each teacher is trained to minimize the cross-entropy loss on its respective domain:

$$\mathcal{L}_T^m = \mathbb{E}_{(x_i^m, y_i^m) \in \mathcal{D}_{tr}^m} \left[ \mathrm{CE}(\boldsymbol{\pi}(x_i^m; \theta_T^m), y_i^m) \right], \tag{1}$$

where $\mathrm{CE}(\cdot, \cdot)$ denotes the cross-entropy between two categorical distributions, $y_i^m$ denotes the ground-truth label of $x_i^m$, and $\boldsymbol{\pi}(x_i^m; \theta_T^m)$ represents the predictive distribution output by the teacher.

A key advantage of this domain-specific training scheme is its computational efficiency. While training $M$ teachers may seem costly, the total computational cost can be significantly less than training a single model on all source domains. This is because each domain-specific teacher can be trained with a smaller batch size (e.g. $1/M$) and requires significantly fewer training steps to become an effective guide for distillation. Crucially, as detailed in Appendix A.1, we empirically find that the student model's performance saturates even before the individual teachers have fully converged, substantially reducing the training overhead that typically makes ensemble methods impractical.

**Complementary Distillation.** As illustrated in Figure 1, DomED employs a complementary distillation strategy designed to transfer the generalization ability of the teacher models to the student. The core principle is to generate distillation targets for samples from a given source domain, $\mathcal{D}_{tr}^m$, using only the teachers that were never trained on it. Specifically, for an input sample $x_i^m \in \mathcal{D}_{tr}^m$, we first gather the output logits $z^n(x_i^m)$ from all complementary teachers $T^n$ (where $n \neq m$). These logits are then aggregated by averaging to produce a single soft target for distillation:

$$\bar{z}(x_i^m) = \frac{1}{M-1} \sum_{\substack{n=1 \\ n \neq m}}^M z^n(x_i^m). \tag{2}$$

The student model is then trained to emulate these aggregated predictions by minimizing a temperature-scaled cross-entropy loss. For each source domain $\mathcal{D}_{tr}^m$, this soft-target loss is defined as:

$$\mathcal{L}_{\mathrm{soft}}^m = \mathbb{E}_{(x_i^m, \cdot) \in \mathcal{D}_{tr}^m} \left[ \mathrm{CE}(\boldsymbol{\pi}(x_i^m; \theta_S), \bar{\boldsymbol{\pi}}(x_i^m; \theta_T, \tau^m)) \right], \tag{3}$$

where $\boldsymbol{\pi}(x_i^m; \theta_S)$ is the distribution predicted by the student model, and $\bar{\boldsymbol{\pi}}(x_i^m; \theta_T, \tau^m)$ is the distribution derived from from the aggregated logits $\bar{z}(x_i^m)$ by applying the softmax function with a temperature hyperparameter $\tau^m$. $\tau^m$ is usually set between 1 and 4, and a larger $\tau$ produces a softer probability distribution. Note that $\bar{\boldsymbol{\pi}}(x_i^m; \theta_T, \tau^m)$ does not receive gradient and $\theta_T$ does not update during distillation. This complementary distillation is also employed in the data allocation scheme shown in Figure 2(d).

As a common practice for knowledge distillation (Hinton et al., 2015), the student model is also trained on the ground-truth labels to achieve higher model accuracy. Similar to the loss in Eq. 1, we minimize the following cross-entropy loss:

$$\mathcal{L}_{\mathrm{hard}}^m = \mathbb{E}_{(x_i^m, y_i^m) \in \mathcal{D}_{tr}^m} \left[ \mathrm{CE}(\boldsymbol{\pi}(x_i^m; \theta_S), y_i^m) \right]. \tag{4}$$

Finally, the two losses in Eqs. 3 and 4 are combined as the training loss of student model:

$$\mathcal{L}_S = \frac{1}{M} \sum_{m=1}^M \left[ \lambda \, \mathcal{L}_{\mathrm{soft}}^m + (1 - \lambda) \mathcal{L}_{\mathrm{hard}}^m \right], \tag{5}$$

where $\lambda$ is a hyperparameter that balances the soft and hard targets. See Appendix F.1 for more details on tuning $\lambda$.

## 2.3 UNCERTAINTY-PRESERVING DISTILLATION

While the distillation scheme discussed in Section 2.2 produces an accurate student model, it discards the ensemble's valuable epistemic uncertainty. This limitation is inherent to the standard cross-entropy objective, which trains the student to match only the *mean* of the teacher predictions. In doing so, it collapses the full predictive distribution into a single point estimate, losing the diversity across teacher outputs that signals model uncertainty. To preserve this crucial information, we distill the ensemble's uncertainty by training the student model to output the parameters of a Dirichlet prior.

**Learning a Dirichlet Prior.**    The Dirichlet distribution is the conjugate prior of the categorical distribution, making it well-suited for modeling the predictive uncertainty of an ensemble. It provides a principled way to represent a distribution over distributions, which naturally captures the set of predictions from the teacher models.

Given $C$ classes, the Dirichlet distribution is defined by its positive concentration parameters $\boldsymbol{\alpha} = [\alpha_1, \cdots, \alpha_C]$. The sum of these parameters, $\alpha_0 = \sum_{c=1}^{C} \alpha_c$, is known as the precision of the distribution; a higher precision indicates lower uncertainty (a more peaked distribution). The probability density function (PDF) is given by:

$$\mathrm{Dir}(\boldsymbol{\pi}; \boldsymbol{\alpha}) = \frac{\Gamma(\alpha_0)}{\prod_{c=1}^{C} \Gamma(\alpha_c)} \prod_{c=1}^{C} (\pi_c)^{\alpha_c - 1}, \tag{6}$$

where $\boldsymbol{\pi}$ is a probability vector. To enable a student model to output this distribution, we parameterize the concentration parameters from its logits, $z_c(x_i^m)$, for an input $x_i^m$ as $\alpha_c(x_i^m) = e^{z_c(x_i^m)}$. The student is then trained to find the Dirichlet parameters that best explain the collection of teacher predictions by minimizing the following negative log-likelihood (NLL) loss (Malinin et al., 2019):

$$\mathcal{L}_{\mathrm{Dir}} = \mathbb{E}_{(x_i^m, \cdot) \in \mathcal{D}_{tr}^m} \left[ \sum_{c=1}^{C} \ln \Gamma\left(\alpha_c\left(x_i^m\right)\right) - \ln \Gamma\left(\alpha_0\left(x_i^m\right)\right) \right.$$

$$\left. - \frac{1}{M-1} \sum_{\substack{n=1 \\ n \neq m}}^{M} \sum_{c=1}^{C} \left(\alpha_c\left(x_i^m\right) - 1\right) \ln \pi_c(x_i^m; \theta_T^n) \right], \tag{7}$$

where $\pi_c(x_i^m; \theta_T^n)$ is the probability assigned to class $c$ by teacher $T^n$.

**Decoupled Uncertainty Distillation.**    While directly minimizing the Dirichlet NLL loss ($\mathcal{L}_{\mathrm{Dir}}$) is a common approach for uncertainty distillation (Malinin & Gales, 2018; Malinin et al., 2019), we find that it can degrade accuracy in the challenging context of domain generalization. This issue arises because $\mathcal{L}_{\mathrm{Dir}}$ implicitly attempts to solve two coupled problems simultaneously: matching the *mean* of the ensemble's predictions and matching their *spread* (i.e., uncertainty). As noted by Ryabinin et al. (2021), these objectives can conflict, forcing the model to compromise on accuracy to better fit the uncertainty.

To resolve this, we propose decoupling these objectives by assigning a specialized loss to each task:

   a) We use the standard distillation loss, $\mathcal{L}_S$, to distill the mean prediction. This cross-entropy-based loss is highly effective at aligning the student with the teachers' mean prediction, thereby preserving model accuracy.
   b) We use the Dirichlet NLL loss, $\mathcal{L}_{\mathrm{Dir}}$, to capture the uncertainty of the ensemble's predictions. When teachers agree, the target Dirichlet distribution is sharp, teaching the student to be confident. When they disagree, the distribution is flat, teaching the student to reflect this uncertainty.

By combining them, we use the dominant $\mathcal{L}_S$ loss to enforce an accurate mean prediction, while $\mathcal{L}_{\mathrm{Dir}}$ focuses on its primary strength: shaping the predictive uncertainty around that mean. This leads to our final, decoupled objective:

$$\mathcal{L}_S' = \mathcal{L}_S + \beta \mathcal{L}_{\mathrm{Dir}}, \tag{8}$$

where $\beta$ is a hyperparameter balancing the two terms. In practice, we set $\beta$ to a small value (e.g., 0.01), effectively using $\mathcal{L}_{\mathrm{Dir}}$ as a regularizer that fine-tunes the model's uncertainty without disrupting the primary accuracy signal from $\mathcal{L}_S$. As we show in our analyses of the loss components (Appendices E and H) and the impact of $\beta$ on convergence (Appendix F.2), this conceptually simple decoupling successfully preserves accuracy while enabling robust uncertainty transfer.

## 3   EXPERIMENTS

Our experiments systematically evaluate DomED on several domain generalization benchmarks. We first systematically compare the performance of six data allocation schemes proposed in this work. We then benchmark DomED against state-of-the-art methods on DomainBed and assess its uncertainty quantification performance using standard metrics. In addition, we present an ablation study of the distillation loss in Appendix E, and verify the architectural robustness of DomED in Appendix I.

Table 1: Comparison of different training data allocation schemes for ensemble learning and knowledge distillation. The best and second-best results indicated by bold and underlined, respectively.

| Dataset | Domain | ERM | DomED (w/o dist.) | Scheme (a) | Scheme (b) | Scheme (c) | Scheme (d) | Scheme (e) | DomED (ours) |
|---|---|---|---|---|---|---|---|---|---|
| **PACS** | *Art Painting* | 84.8±0.2 | 78.2±2.8 | 85.9±0.5 | 86.2±1.3 | 87.2±0.5 | 87.5±0.8 | 85.9±0.6 | **87.5**±0.5 |
| | *Cartoon* | 80.0±0.6 | 69.2±1.0 | 81.7±0.7 | 80.4±0.8 | 81.0±0.6 | **82.1**±0.5 | 80.1±0.7 | 81.5±0.3 |
| | *Photo* | 96.2±0.0 | 95.0±0.2 | 96.9±0.3 | 96.7±0.2 | 97.0±0.2 | 96.5±0.8 | 96.4±0.2 | **97.1**±0.2 |
| | *Sketch* | 79.3±0.3 | 68.6±4.8 | 79.7±0.4 | 79.1±1.1 | 81.4±0.4 | 80.6±0.2 | 77.3±0.8 | **81.6**±0.4 |
| | *Avg.* | 85.1±0.2 | 77.7±0.9 | 86.1±0.4 | 85.6±0.4 | 86.6±0.3 | 86.7±0.2 | 84.9±0.3 | **86.9**±0.2 |
| **Office Home** | *Art* | 61.3±0.3 | 64.6±0.4 | 67.2±0.5 | 67.3±0.6 | 67.5±0.2 | 67.6±0.2 | 67.4±0.3 | **67.7**±0.1 |
| | *Clipart* | 52.1±0.4 | 52.1±0.2 | 56.4±0.6 | 56.4±0.6 | 56.7±0.3 | 56.9±0.2 | 56.7±0.3 | **57.1**±0.2 |
| | *Product* | 76.6±0.3 | 75.6±0.3 | 77.7±0.2 | 77.6±0.3 | 78.1±0.2 | 78.0±0.2 | 77.7±0.1 | **78.2**±0.2 |
| | *RealWorld* | 78.5±0.2 | 79.1±0.2 | 81.3±0.1 | 81.5±0.2 | 81.7±0.1 | 81.6±0.1 | 81.2±0.1 | **81.8**±0.1 |
| | *Avg.* | 67.1±0.2 | 67.8±0.2 | 70.6±0.3 | 70.6±0.4 | 71.0±0.2 | 71.0±0.1 | 70.7±0.2 | **71.2**±0.2 |
| **VLCS** | *Caltech101* | 97.8±0.1 | 93.8±3.7 | 98.2±0.3 | 98.3±0.3 | **98.5**±0.2 | 98.2±0.1 | 97.8±0.4 | **98.5**±0.1 |
| | *LabelMe* | 64.2±0.3 | 58.0±1.5 | 65.4±0.4 | **65.5**±0.5 | 64.8±0.3 | 64.9±0.2 | 63.6±0.4 | 65.3±0.2 |
| | *SUN09* | 72.6±0.4 | 77.0±0.9 | 73.3±0.9 | 74.8±0.4 | 78.0±0.2 | **78.8**±0.2 | 72.9±0.3 | 78.4±0.3 |
| | *VOC2007* | 77.6±0.4 | 76.7±1.5 | 77.3±0.4 | 77.7±0.3 | **78.9**±0.2 | 77.8±0.1 | 77.2±0.4 | 78.4±0.2 |
| | *Avg.* | 78.0±0.3 | 76.4±1.7 | 78.5±0.5 | 79.1±0.3 | **80.1**±0.2 | 79.9±0.1 | 77.9±0.4 | **80.1**±0.1 |

## 3.1 EXPERIMENTAL SETUP

**Datasets and evaluation protocol.** We use the following datasets in our experiments: 1) PACS (Li et al., 2017), a widely used multi-source domain generalization dataset comprising 9,991 images across 7 classes and 4 domains (Art_painting, Cartoon, Photo, and Sketch), 2) Office-Home (Venkateswara et al., 2017), a dataset of 15,500 images from 65 classes and 4 domains (Art, Clipart, Product, and Real World), 3) VLCS (Fang et al., 2013), which contains 10,729 images of 5 classes sourced from Caltech101, LabelMe, SUN09, and VOC 2007, 4) TerraIncognita (Beery et al., 2018), a slightly larger dataset of 24,788 images from 10 classes and 4 domains, and 5) DomainNet (Peng et al., 2019), a significantly larger dataset of 586,575 images from 345 classes and 6 domains. Following the evaluation protocol of DomainBed (Gulrajani & Lopez-Paz, 2020), we perform leave-one-domain-out evaluation and use training-domain validation for hyperparameter tuning. In Tables 1 and 2, the reported accuracies are the average of 7 runs.

**Implementation Details.** To ensure a fair comparison, we follow the standard protocol of DomainBed (Gulrajani & Lopez-Paz, 2020) for all experiments. We use a pre-trained ResNet-50 (He et al., 2016) backbone. For the PACS, TerraIncognita, and DomainNet datasets, we use the Adam optimizer (Kingma & Ba, 2014) with a learning rate of $5 \times 10^{-5}$. For OfficeHome and VLCS, we use stochastic gradient descent (SGD) with an initial learning rate of $2 \times 10^{-2}$ and a momentum of 0.9. Other hyperparameters, such as batch size, dropout rate, and weight decay, are adopted from recent state-of-the-art methods (Cha et al., 2021; 2022).

## 3.2 DOMAIN GENERALIZATION

**Data allocation schemes.** In addition to the complementary data allocation scheme illustrated in Figure 1, we investigate five other schemes as shown in Figure 2. Among them, scheme (a) is simple self-distillation without ensembling, scheme (b) corresponds to regular ensembling and distillation that use the full training data for both purposes, and the other three can be considered as variants of DomED. Compared to DomED, scheme (c) distills from all teachers regardless of their training domains, scheme (d) allocates each teacher $M - 1$ training domains instead of 1 but also distills on complementary domains, whereas scheme (e) distills on the respective training domains of teachers.

We evaluate the six data allocation schemes along with two baselines, ERM and DomED without distillation, on PACS, OfficeHome, and VLCS. The results are shown in Table 1. Comparing schemes (b) and (c), the latter achieves better accuracies on all three datasets, which suggests that training base models on different subsets of domains can indeed lead to better generalization, possibly due to increased model diversity. By comparing DomED with scheme (c), we also observe that

Table 2: Classification accuracy (%) on the DomainBed benchmark. Methods are grouped into standalone approaches and those using weight averaging. Best and second-best results in each group are in **bold** and underlined, respectively. "TerraInc." stands for TerraIncognita. [†]Results reported by Gulrajani & Lopez-Paz (2020). [‡]Results reported by Di Zhao et al. (2025).[§]Test-time ensembling. [¶]Weight averaging from 60 runs.

| Method | PACS | OfficeHome | VLCS | TerraInc. | DomainNet | Avg. |
|---|---|---|---|---|---|---|
| ERM (Vapnik, 1999) | $85.1_{\pm 0.2}$ | $67.1_{\pm 0.2}$ | $78.0_{\pm 0.3}$ | $47.8_{\pm 0.6}$ | $44.0_{\pm 0.1}$ | 64.4 |
| CORAL[†] (Sun & Saenko, 2016) | $86.2_{\pm 0.3}$ | $68.7_{\pm 0.3}$ | $78.8_{\pm 0.6}$ | $47.6_{\pm 1.0}$ | $41.5_{\pm 0.1}$ | 64.5 |
| DANN[†] (Ganin et al., 2016) | $83.6_{\pm 0.4}$ | $65.9_{\pm 0.6}$ | $78.6_{\pm 0.4}$ | $46.7_{\pm 0.5}$ | $38.3_{\pm 0.1}$ | 62.6 |
| MMD[†] (Li et al., 2018b) | $84.7_{\pm 0.5}$ | $66.3_{\pm 0.1}$ | $77.5_{\pm 0.9}$ | $42.2_{\pm 1.6}$ | $23.4_{\pm 9.5}$ | 58.8 |
| IRM[†] (Arjovsky et al., 2019) | $83.5_{\pm 0.8}$ | $64.3_{\pm 2.2}$ | $78.5_{\pm 0.5}$ | $47.6_{\pm 0.8}$ | $33.9_{\pm 2.8}$ | 61.1 |
| Fish (Shi et al., 2021) | $85.5_{\pm 0.3}$ | $68.6_{\pm 0.4}$ | $77.8_{\pm 0.3}$ | $45.1_{\pm 1.3}$ | $42.7_{\pm 0.2}$ | 63.9 |
| SagNet[†] (Nam et al., 2021b) | $86.3_{\pm 0.2}$ | $68.1_{\pm 0.1}$ | $77.8_{\pm 0.5}$ | $48.6_{\pm 1.0}$ | $40.3_{\pm 0.1}$ | 64.2 |
| SelfReg (Kim et al., 2021) | $85.6_{\pm 0.4}$ | $67.9_{\pm 0.7}$ | $77.8_{\pm 0.9}$ | $47.0_{\pm 0.3}$ | $42.8_{\pm 0.1}$ | 64.2 |
| NKD[‡] (Wang et al., 2021) | $83.3_{\pm 0.4}$ | $71.1_{\pm 0.3}$ | $77.1_{\pm 0.3}$ | $37.2_{\pm 0.3}$ | $42.4_{\pm 0.2}$ | 62.2 |
| MIRO (Cha et al., 2022) | $85.4_{\pm 0.4}$ | $70.5_{\pm 0.4}$ | $79.0_{\pm 0.0}$ | $50.4_{\pm 1.1}$ | $44.3_{\pm 0.2}$ | 65.9 |
| KDDRL (Niu et al., 2023) | $86.6_{\pm 0.8}$ | $66.9_{\pm 1.6}$ | $77.3_{\pm 0.5}$ | $48.0_{\pm 1.1}$ | $38.5_{\pm 0.3}$ | 63.4 |
| SAGM (Wang et al., 2023b) | $86.6_{\pm 0.2}$ | $70.1_{\pm 0.2}$ | $\underline{80.0}_{\pm 0.3}$ | $48.8_{\pm 0.9}$ | $45.0_{\pm 0.2}$ | 66.1 |
| DomainDrop (Guo et al., 2023) | $\mathbf{87.9}_{\pm 0.3}$ | $68.7_{\pm 0.1}$ | $79.8_{\pm 0.3}$ | $\mathbf{51.5}_{\pm 0.4}$ | $44.4_{\pm 0.5}$ | $\underline{66.5}$ |
| RISE[‡] (Huang et al., 2023) | $85.0_{\pm 0.3}$ | $71.5_{\pm 0.2}$ | $77.6_{\pm 0.2}$ | $39.0_{\pm 0.3}$ | $45.2_{\pm 0.2}$ | 63.7 |
| GMDG (Tan et al., 2024) | $85.6_{\pm 0.3}$ | $70.7_{\pm 0.2}$ | $79.2_{\pm 0.3}$ | $\underline{51.1}_{\pm 0.9}$ | $44.6_{\pm 0.1}$ | 66.3 |
| XDomainMix (Liu et al., 2024) | $86.4_{\pm 0.4}$ | - | - | $48.2_{\pm 1.3}$ | $44.4_{\pm 0.2}$ | - |
| Arith (Wang et al., 2025) | $86.5_{\pm 0.3}$ | $69.4_{\pm 0.1}$ | $79.4_{\pm 0.3}$ | $48.1_{\pm 1.2}$ | $41.5_{\pm 0.1}$ | 65.0 |
| GGA (Ballas & Diou, 2025) | $86.4_{\pm 0.5}$ | $67.0_{\pm 0.3}$ | $78.7_{\pm 0.8}$ | $48.5_{\pm 1.1}$ | $44.4_{\pm 0.2}$ | 65.0 |
| BOLD[‡] (Di Zhao et al., 2025) | $85.7_{\pm 0.2}$ | $\mathbf{72.6}_{\pm 0.2}$ | $78.7_{\pm 0.2}$ | $44.3_{\pm 0.3}$ | $\mathbf{46.9}_{\pm 0.2}$ | 65.6 |
| **DomED (ours)** | $\underline{86.9}_{\pm 0.2}$ | $\underline{71.2}_{\pm 0.2}$ | $\mathbf{80.1}_{\pm 0.1}$ | $50.2_{\pm 0.5}$ | $\underline{45.7}_{\pm 0.1}$ | $\mathbf{66.8}$ |
| SWAD (Cha et al., 2021) | $88.1_{\pm 0.1}$ | $70.6_{\pm 0.2}$ | $79.1_{\pm 0.1}$ | $50.0_{\pm 0.3}$ | $46.5_{\pm 0.1}$ | 66.9 |
| CORAL (w/ SWAD) (Cha et al., 2021) | $88.3_{\pm 0.1}$ | $71.3_{\pm 0.1}$ | $78.9_{\pm 0.1}$ | $51.0_{\pm 0.1}$ | $46.8_{\pm 0.0}$ | 67.3 |
| SAM (w/ SWAD) (Cha et al., 2021) | $87.1_{\pm 0.2}$ | $69.9_{\pm 0.2}$ | $78.5_{\pm 0.1}$ | $45.3_{\pm 0.9}$ | $46.5_{\pm 0.1}$ | 65.5 |
| DNA (w/ SWAD) (Chu et al., 2022) | $88.4_{\pm 0.1}$ | $71.2_{\pm 0.1}$ | $79.0_{\pm 0.1}$ | $\underline{52.2}_{\pm 0.4}$ | $\underline{47.2}_{\pm 0.1}$ | 67.6 |
| EoA[§] (Arpit et al., 2022) | $\underline{88.6}$ | $\mathbf{72.5}$ | $79.1$ | $\mathbf{52.3}$ | $\mathbf{47.4}$ | $\mathbf{68.0}$ |
| DiWA[¶] (Rame et al., 2022) | $\mathbf{89.0}$ | $\underline{71.6}$ | $\underline{79.4}$ | $49.0$ | $46.3$ | 67.1 |
| **DomED (w/ SWAD) (ours)** | $88.4_{\pm 0.2}$ | $71.6_{\pm 0.2}$ | $\mathbf{80.2}_{\pm 0.1}$ | $51.9_{\pm 0.3}$ | $46.7_{\pm 0.2}$ | $\underline{67.8}$ |

complementary distillation achieves comparable or slightly better accuracies than distillation on all domains, thus corroborating the intuition that DomED can distill the generalization ability into student model. Moreover, despite their similar classification performance, DomED outperforms scheme (c) in terms of uncertainty quantification (see Table 3). In contrast, distillation only on training domains (scheme (e)) can result in significantly worse accuracies compared to complementary distillation (scheme (d)). Furthermore, while both DomED and scheme (d) employ complementary distillation with comparable accuracies, DomED restricts each teacher to a single domain. This design minimizes training overhead while maximizing model diversity (see Appendix B.1), a prerequisite for robust uncertainty quantification. Finally, we note that the teacher models of DomED are relatively weak as each individual model is trained on only one domain; thus, they perform poorly compared to the distilled model. Nevertheless, the distilled model proves to be resilient even when individual teachers are trained on scarce data, mitigating the risk of negative transfer (see Appendix C). Collectively, the systematic evaluation confirms that our data allocation strategy is a principled and effective design choice for achieving high teacher diversity and robust distillation.

**Results on DomainBed.** We evaluate DomED on the DomainBed benchmark against a wide array of state-of-the-art methods, with results presented in Table 2. To provide a clear comparison, we group methods into standard approaches and those that utilize weight averaging techniques like SWAD to further boost performance. As a standalone method, DomED demonstrates strong performance, achieving significant gains over the ERM baseline: +1.8 pp on PACS, +4.1 pp on OfficeHome, +2.1 pp on VLCS, +2.4 pp on TerraIncognita, and +1.7 pp on DomainNet. The improvement is particularly pronounced on OfficeHome, a dataset known for its large inter-domain gap. This setting is ideal for our approach, as it allows domain-specific teachers to become highly specialized, and our complementary distillation can then effectively integrate their diverse knowledge to improve generalization. When paired with SWAD, our method achieves a top-tier average accuracy of 67.8%,

which is highly competitive with other leading methods that employ significantly more computation, such as EoA (test-time ensembling) and DiWA (weight averaging from 60 runs). This highlights the efficiency of our approach, which delivers state-of-the-art performance without significant overhead. See Appendix A for a detailed analysis of computational cost.

## 3.3 UNCERTAINTY QUANTIFICATION

Beyond strong generalization performance, a crucial advantage of DomED is its ability to provide reliable uncertainty estimates. To evaluate this, we employ four standard metrics: mean classification error (ERR), prediction rejection ratio (PRR) (Malinin et al., 2019), expected calibration error (ECE) (Guo et al., 2017), and negative log-likelihood (NLL). A high PRR is desirable, as it indicates the model can effectively detect and reject its own incorrect predictions. A low ECE signifies well-calibrated confidence. They both are essential for trustworthy OOD generalization.

**Calibration and Reliability Analysis.** We conduct a detailed analysis on the PACS dataset, comparing DomED to various baselines. For reference, we also include two full test-time ensembles: a standard ensemble (3 base models) and an EoA ensemble (Arpit et al., 2022). The results are shown in Table 3 (see Appendix J for more detailed results). Standard methods like ERM and CORAL achieve reasonable accuracy but are poorly calibrated, evidenced by high ECE and NLL values. The poor calibration stems from a rapid entropy collapse during training, whereas the distillation process of DomED retains the dark knowledge from diverse teacher models, resulting in high-entropy predictions (see Appendix D). Post-hoc temperature scaling (Guo et al., 2017) improves calibration but fails to enhance the model's ability to reject incorrect predictions (PRR). Approximate Bayesian methods like MC Dropout (Gal & Ghahramani, 2016), which require multiple forward passes at inference, prove ineffective in this DG context, degrading accuracy without improving calibration. Furthermore, $EnD^2$, while designed for uncertainty distillation, performs poorly due to the absence of ground-truth labels in the distillation process. In contrast, DomED strikes an exceptional balance between accuracy and calibration. See Appendix G for a comparison with more calibration strategies. The analysis highlights two key aspects of its design. First, the raw 'DomED Teachers' ensemble performs poorly on its own, confirming that our distillation process is essential to effectively integrate the specialists' knowledge.

Table 3: Uncertainty quantification performance on PACS. DomED is the best single-inference method.

| Model | ERR↓ | ECE↓ | NLL↓ | PRR↑ |
|---|---|---|---|---|
| Ensemble | 0.130 | 0.038 | 0.471 | 0.798 |
| EoA | 0.114 | 0.058 | 0.403 | 0.837 |
| ERM | 0.149 | 0.089 | 0.625 | 0.776 |
| Temp. Scaling | 0.147 | 0.048 | 0.487 | 0.773 |
| MC Drop (p=0.5) | 0.175 | 0.129 | 0.873 | 0.746 |
| MC Drop (p=0.1) | 0.157 | 0.112 | 0.755 | 0.769 |
| CORAL | 0.138 | 0.082 | 0.586 | 0.598 |
| $EnD^2$ | 0.326 | 0.220 | 1.580 | 0.084 |
| DomED Teachers | 0.281 | 0.148 | 1.063 | 0.536 |
| Scheme (c) | 0.134 | 0.054 | 0.502 | 0.751 |
| Scheme (d) | 0.133 | 0.091 | 0.523 | 0.743 |
| **DomED (Ours)** | **0.131** | **0.044** | **0.473** | **0.787** |

Table 4: ROC-AUC of OOD detection. T.Unc and K.Unc refer to total and knowledge uncertainty, respectively.

| Models | PACS | | OfficeHome | |
|---|---|---|---|---|
| | T.Unc | K.Unc | T.Unc | K.Unc |
| ERM | 0.71 | – | 0.67 | – |
| Ensemble | **0.73** | 0.71 | **0.69** | 0.70 |
| DomED | 0.71 | **0.72** | 0.60 | **0.70** |

| Models | VLCS | | TerraIncognita | |
|---|---|---|---|---|
| | T.Unc | K.Unc | T.Unc | K.Unc |
| ERM | 0.51 | – | 0.75 | – |
| Ensemble | 0.53 | 0.54 | **0.78** | **0.79** |
| DomED | **0.57** | **0.69** | 0.77 | 0.75 |

Second, DomED outperforms alternative distillation schemes (c) and (d) across all metrics, demonstrating the superiority of the complementary distillation strategy. Overall, DomED achieves the best performance among all single-inference methods. Crucially, while test-time ensembles serve as a strong performance ceiling, the competitive performance of DomED demonstrates the efficacy of our distillation strategy in transferring the benefits of an ensemble into a single efficient model. Finally, we present additional results on the impact of teacher diversity on model performance in Appendix B.

**Out-of-Distribution Detection.** We further assess the utility of DomED's uncertainty estimates for out-of-distribution (OOD) detection. Following Malinin et al. (2019), we treat samples from the unseen test domain as OOD and samples from the source domains as in-distribution (ID). We use the area under the receiver operating characteristic curve (ROC-AUC) to measure the model's ability to distinguish between these two groups using total uncertainty and knowledge uncertainty. As shown in Table 4 (see Appendix J for more detailed results), DomED's OOD detection performance is highly competitive with that of a full test-time ensemble (3 base models). On average, it closely matches the

ensemble's ability to identify novel, unseen domains, which is crucial for safe and reliable real-world deployment.

## 4 RELATED WORK

### 4.1 DOMAIN GENERALIZATION

Domain Generalization (DG) addresses the challenge of dataset shift (Moreno-Torres et al., 2012), where a model's performance degrades when training and testing distributions differ. Existing DG methods are often categorized into three main groups (Wang et al., 2022): data manipulation, representation learning, and learning strategies. Data manipulation methods augment training data to enhance diversity and quantity (Volpi et al., 2018; Zhou et al., 2021). Representation learning aims to find domain-invariant features, often through feature alignment or disentanglement (Du et al., 2021; Mahajan et al., 2021; Zhang et al., 2022; Dayal et al., 2024). Recent work in this area has explored finer-grained control, such as by suppressing domain-sensitive channels (Guo et al., 2023) or establishing more general objectives for invariant features (Tan et al., 2024). Finally, various learning strategies adapt existing machine learning techniques for DG, including ensemble learning (Arpit et al., 2022; Lee et al., 2022; Niu et al., 2023), meta-learning (Li et al., 2018a; Guan et al., 2023; Chen et al., 2023; Wang et al., 2025), and gradient operations (Shi et al., 2021; Ballas & Diou, 2025).

### 4.2 ENSEMBLE METHODS AND DISTILLATION

Ensemble methods have proven effective in improving OOD generalization. For instance, Arpit et al. (2022) employs an ensemble of moving average models to achieve better generalization performance, which, however, relies on costly test-time ensembling that incurs significant inference overhead. To avoid test-time ensembling, efficient fusion techniques have also been developed, including knowledge distillation (Hinton et al., 2015), weight averaging (Cha et al., 2021; Chu et al., 2022; Rame et al., 2022), and model merging (Ding et al., 2025). In the context of domain generalization, knowledge distillation can be enhanced by gradient regularization (Wang et al., 2021) or combined with language guidance (Huang et al., 2023). Apart from full-model ensembling, parameter-efficient expert aggregation has also attracted attention. For instance, AdapterFusion (Pfeiffer et al., 2021) leverages knowledge from multiple tasks by training and fusing task-specific adapters, and recent approaches focus on merging Low-Rank Adaptation (LoRA) modules (Hu et al., 2022) with minimal interference (Yadav et al., 2023). In contrast to these parameter-space techniques that primarily focus on accuracy, DomED operates in the output space to explicitly transfer both generalization capabilities and epistemic uncertainty.

Theoretical work has shown that given the "multi-view" structure of data, an ensemble of independently trained neural networks can provably improve test accuracy, and such capability can be provably distilled into a single model (Allen-Zhu & Li, 2020). For domain generalization, Zhou et al. (2021) introduce domain adaptive ensemble learning, which encompasses a shared CNN feature extractor and multiple classifier heads, each excelling in one specific domain. In a similar vein, Niu et al. (2023) also use a shared feature extractor with multiple domain-specific heads, followed by a two-stage distillation scheme. However, the complex scheme yields only marginal gains on domain generalization tasks. Furthermore, Di Zhao et al. (2025) apply the idea of domain-specific experts to distillation from large pretrained models. Yao et al. (2023) propose to dynamically weight domain-specific models based on domain relations during inference. Other unconventional approaches include ensembling logits with the same label but from different domains (Lee et al., 2022) and ensembling meta-source models (Yan & Guo, 2025), though the latter leverages test-time batch statistics, positioning it closer to test-time adaptation than pure domain generalization. In addition, ensemble distillation has been utilized for out-of-distribution (OOD) detection (Wang et al., 2023a), a different focus than generalization to unseen domains. Despite this body of work, a systematic study of how the data allocation scheme for ensemble learning and distillation can be tailored to the specific setting of domain generalization has been largely overlooked.

### 4.3 UNCERTAINTY ESTIMATION

In machine learning applications, distinguishing between data (aleatoric) uncertainty and model (epistemic) uncertainty is essential to understand the decisions or predictions made by models (Gal, 2016). Ensemble methods are particularly effective at separating and quantifying these two types of uncer-

tainty (Lakshminarayanan et al., 2017). An intuitive approach to uncertainty quantification (Hüllermeier & Waegeman, 2021; Depeweg et al., 2017) is to first derive the total uncertainty as the entropy of the expected predictive distribution for a data point $(x, y)$, i.e. $U_{tot} = \mathcal{H}(\mathbb{E}_{p(\theta|\mathcal{D})}[p_\theta(y|x)]$, where $\theta$ denotes model parameters, and $\mathcal{D}$ represents the training data. Similarly, data uncertainty can be defined as the expected entropy of the predictive distribution of individual models, i.e. $U_{ale} = \mathbb{E}_{p(\theta|\mathcal{D})}[\mathcal{H}(p_\theta(y|x)]$; and model uncertainty can be naturally defined as $U_{tot} - U_{ale}$.

The Dirichlet distribution plays a crucial role in uncertainty quantification for classification models, particularly when distilling the knowledge of an ensemble into a single, efficient student model. As the conjugate prior of the categorical distribution, it provides a principled way to model a distribution over predicted probabilities. This approach is central to methods like Dirichlet prior networks (Tsiligkaridis, 2021; Malinin & Gales, 2018; Malinin et al., 2019) and evidential learning (Joo et al., 2020; Dawood et al., 2023; Schreck et al., 2024). However, training a model by directly minimizing the Dirichlet negative log-likelihood (NLL) can be problematic, as the loss function often degrades model accuracy in the process of improving uncertainty estimates (Ryabinin et al., 2021).

## 5 CONCLUSION

In this work, we introduced DomED, a redesigned ensemble distillation framework for domain generalization that jointly addresses model accuracy and uncertainty quantification. By combining a specialized data allocation strategy to enhance teacher diversity with a novel technique for decoupling mean and uncertainty distillation, DomED successfully transfers the knowledge of an ensemble into a single, efficient model. Our extensive experiments demonstrate that DomED achieves competitive accuracy while providing uncertainty estimates that rival a full, computationally expensive ensemble, underscoring the importance and feasibility of jointly optimizing for both objectives.

While our approach is more efficient than standard ensembling, its computational cost remains a limitation compared to single-model training, presenting a clear direction for future work on improving efficiency. Furthermore, the core principle of decoupled uncertainty distillation is broadly applicable. Extending this framework to other tasks, such as uncertainty-aware regression or dense prediction, is an exciting avenue for future research. Overall, this work provides a robust and practical foundation for building more reliable and uncertainty-aware models for domain generalization.

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

## A  COMPUTATIONAL COST ANALYSIS

### A.1  OVERHEAD OF TRAINING TEACHER MODELS

A key practical advantage of DomED is its computational efficiency, despite using an ensemble of teacher models. This section demonstrates that the full training of teacher models is unnecessary for effective distillation, which significantly reduces the method's computational cost. As shown in Figure 3, the distilled student models' performance converges and stabilizes early in the training process (within approximately 1,000 steps), long before the teacher models have fully converged on their respective source domains. This observation indicates that while the guidance of teacher models is essential, the student's performance is not sensitive to the teachers' final, fully converged state.

This observation significantly reduces the computational cost of our method. For a training set with $M$ domains, we can limit the training steps for each teacher to a fraction (e.g., $1/M$) of the steps used for a standard single-model baseline.[1] Consequently, the total steps of training $M$ teachers are comparable to training a single model on all domains, making DomED a computationally practical approach.

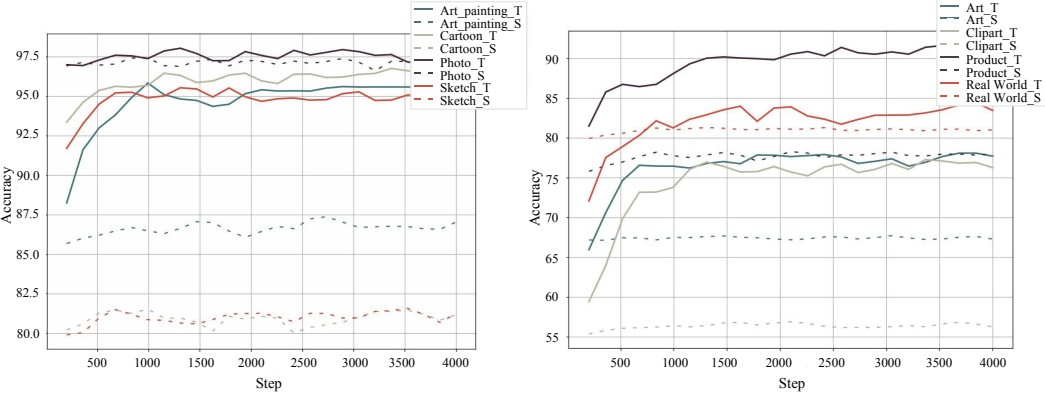

Figure 3: Impact of teacher training steps on student performance for (left) PACS and (right) OfficeHome. Solid lines represent teacher accuracy on their source domain, while dashed lines show the distilled student's accuracy on the corresponding target domain. Note that student performance (dashed lines) converges and stabilizes early, long before the teacher models (solid lines) have fully converged.

### A.2  TRAINING AND DISTILLATION COST

Table 5: Computational cost and performance comparison on PACS (Target domain: Sketch).

| Method | Wall-Time (h) | Peak Mem (GB) ↓ | Acc. (%) ↑ | ECE ↓ | NLL ↓ |
|---|---|---|---|---|---|
| ERM | **1.91** | 8.13 | 79.3 | 0.121 | 0.919 |
| DNA (Chu et al., 2022) | 3.23 | 8.17 | 79.8 | 0.087 | 0.708 |
| **DomED** | 2.86 | 8.13 | **81.6** | **0.028** | **0.609** |
| ↪ *DomED (Teachers)* | *0.78* | ***3.00*** | - | - | - |
| ↪ *DomED (Student)* | *2.08* | *8.13* | - | - | - |

We provide a detailed computational cost analysis on PACS in Table 5. The total training time for DomED is around 50% longer than ERM while achieving significantly better accuracy and calibration. The training of the teacher ensemble is particularly fast (0.78 h) and does not scale linearly with the number of domains. This efficiency stems from two factors: 1) Since each teacher sees only one domain, it is trained with a batch size $1/M$ of the standard ERM batch size, reducing the cost of each forward/backward pass. 2) As discussed in Appendix A.1, effective distillation does not require

Table 6: Total training time comparison with SWAD on PACS (Target: Sketch).

| Method | Wall-Time(h)↓ |
|---|---|
| SWAD | 2.41 |
| DomED + SWAD | 3.27 |

---

[1]According to the DomainBed protocol, the number of training steps is 5,000 for PACS, OfficeHome, VLCS, TerraIncognita, and 15,000 for DomainNet.

fully converged teachers because the student model saturates early, which allows us to significantly reduce the number of teacher training steps compared to a standard run.

When combined with SWAD (Cha et al., 2021), we apply weight averaging only during the distillation phase of the student model. As shown in Table 6, this incurs a small overhead compared to standard SWAD, yet achieves higher accuracy by combining the benefits of ensemble knowledge and flat-minima optimization (See Table 2).

# B ANALYSIS OF TEACHER DIVERSITY

## B.1 VISUALIZING TEACHER DIVERSITY

A central design principle of DomED is the use of a domain-specific data allocation scheme to foster the diversity required for effective ensemble learning. This section validates this strategy by visually analyzing the performance of teachers trained on distinct, single domains against those trained on dual-domain combinations. The accuracy heatmaps in Figures 4, 5, 6, and 7 visualize this comparison across the PACS, OfficeHome, VLCS, and TerraIncognita datasets, respectively.

We observe a consistent pattern across these benchmarks where single-domain teachers (left panels) exhibit significantly higher specialization compared to their dual-domain counterparts (right panels). For instance, in the PACS dataset (Figure 4), a teacher trained exclusively on the "Art Painting" domain excels in its source domain but exhibits higher variance on others. In contrast, dual-domain teachers show more uniform performance across test domains. This empirical evidence supports our design choice: training each teacher on a single, distinct domain effectively encourages the development of specialized representations, which is a prerequisite for maximizing ensemble diversity.

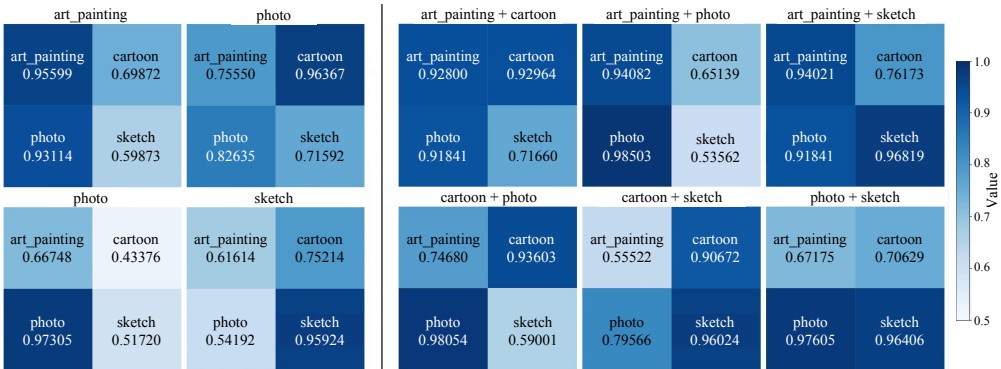

Figure 4: Accuracy heatmaps comparing single-domain (left) and dual-domain (right) teachers on the **PACS** dataset. Each block represents a trained teacher model, and each patch represents a test domain. The more varied color patterns on the left indicate higher specialization and diversity.

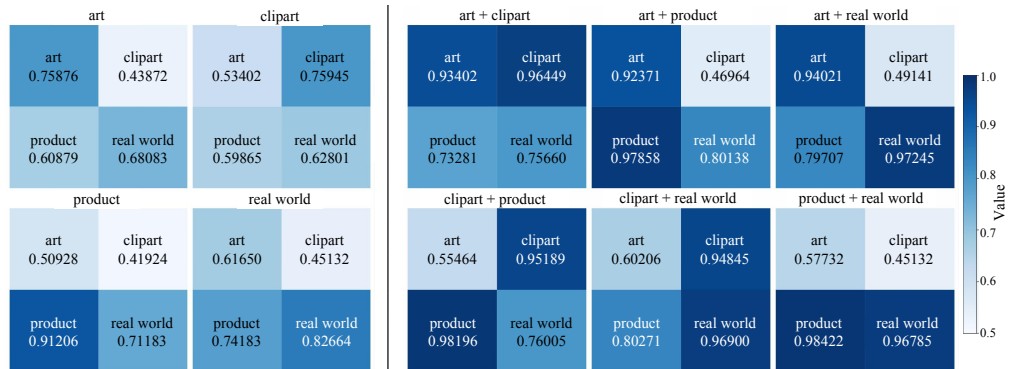

Figure 5: Accuracy heatmaps for single-domain (left) and dual-domain (right) teachers trained on the **OfficeHome** dataset.

The heatmaps further reveal how diversity varies across datasets. On the OfficeHome dataset (Figure 5), single-domain teachers maintain reasonable accuracy on unseen domains. This indicates

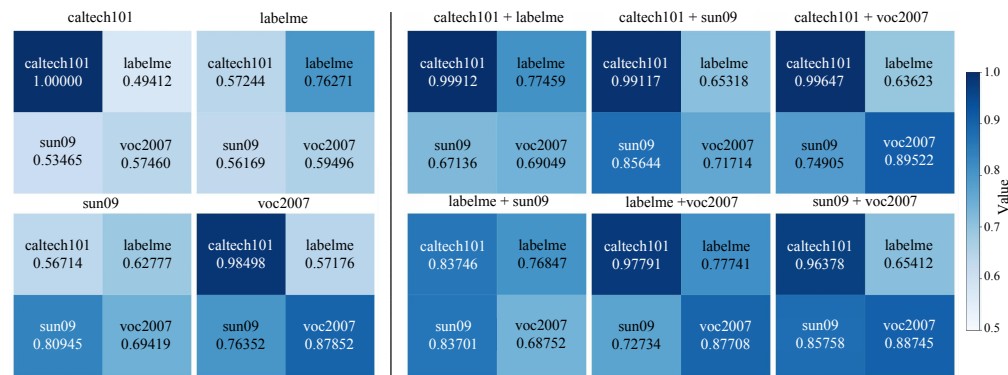

Figure 6: Accuracy heatmaps for single-domain (left) and dual-domain (right) teachers trained on the **VLCS** dataset.

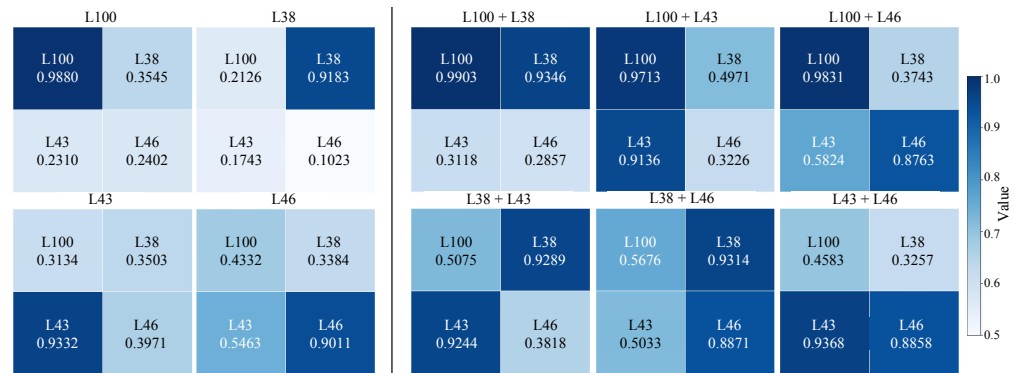

Figure 7: Accuracy heatmaps for single-domain (left) and dual-domain (right) teachers on the **TerraIncognita** dataset. Single-domain teachers exhibit high specialization.

that despite the stylistic differences between domains (e.g., Art, Product), the teachers successfully learn transferable semantic features that enable effective cross-domain generalization. In contrast, we observe a different behavior on the TerraIncognita dataset. As shown in Figure 7, single-domain teachers on TerraIncognita exhibit a distinct pattern of high specialization, achieving high accuracy on their respective source domains but suffering significant performance drops on unseen domains. We hypothesize that this behavior stems from the nature of camera trap data, where static backgrounds (e.g., specific vegetation or terrain) encourage models to learn domain-specific shortcuts that do not transfer well. Despite this challenging scenario where teacher transferability is limited, DomED effectively recovers useful signals to outperform the ERM baseline (50.2% vs. 47.8%).

### B.2  IMPACT OF TEACHER DIVERSITY ON STUDENT PERFORMANCE

This section quantifies the impact of teacher diversity on student performance through a controlled experiment on the PACS dataset. We compared our proposed single-domain allocation strategy (high granularity, high diversity) against a dual-domain allocation baseline (low granularity, low diversity) where teachers are trained on pairs of domains. As presented in Table 7, the results demonstrate that the higher diversity provided by single-domain teachers translates into superior student performance. While accuracy improves by 2.0%, the impact on uncertainty is most notable: the single-domain strategy reduces ECE by over 50% (from 0.107 to 0.044) compared to the dual-domain baseline.

### B.3  IMPACT OF ENSEMBLE SIZE ON STUDENT PERFORMANCE

Since our method ties the number of domain-specific teachers (or ensemble size) to that of training domains, varying the latter can affect teacher diversity as a result. To quantify the impact of ensemble size, we conducted an ablation study on the PACS dataset. We systematically train models on all possible 2-domain subsets ($M = 2$, each resulting in 2 domain-specific teachers) and compare their average performance against the standard 3-domain configuration ($M = 3$) as well as an ERM baseline trained on the same 2-domain subsets.

Table 7: Impact of teacher diversity: Comparison between single-domain (DomED) and dual-domain allocation strategies on PACS.

| Target Domain | Allocation | Diversity | Acc. (%)↑ | ECE↓ | NLL↓ | PRR↑ |
|---|---|---|---|---|---|---|
| Art Painting | Dual-domain | Low | 85.1 | 0.109 | 0.521 | 0.772 |
| | **Single-domain (DomED)** | **High** | **87.5** | **0.049** | **0.497** | **0.781** |
| Cartoon | Dual-domain | Low | 78.3 | 0.076 | 0.726 | 0.654 |
| | **Single-domain (DomED)** | **High** | **81.5** | **0.038** | **0.628** | **0.738** |
| Photo | Dual-domain | Low | 96.0 | 0.094 | 0.200 | 0.882 |
| | **Single-domain (DomED)** | **High** | **97.1** | **0.061** | **0.158** | **0.936** |
| Sketch | Dual-domain | Low | 80.2 | 0.150 | 0.716 | 0.614 |
| | **Single-domain (DomED)** | **High** | **81.6** | **0.028** | **0.609** | **0.694** |
| **Average** | Dual-domain | Low | 84.9 | 0.107 | 0.541 | 0.730 |
| | **Single-domain (DomED)** | **High** | **86.9** | **0.044** | **0.473** | **0.787** |

The results, summarized in Table 8, reveal that despite the reduction in teachers, DomED ($M = 2$) still outperforms the ERM baseline trained on the identical data subset. While accuracy is comparable, DomED maintains significantly better calibration (ECE 0.050 vs. 0.097). When compared to the full ensemble, reducing the size

Table 8: Impact of ensemble size ($M$) on student performance evaluated on PACS.

| Metric | DomED ($M = 3$) | DomED ($M = 2$) | ERM ($M = 2$) |
|---|---|---|---|
| **Accuracy** ↑ | 86.9% ±0.2 | 84.6% ±0.3 | 84.3% ±0.2 |
| **ECE** ↓ | 0.044 | 0.050 | 0.097 |
| **NLL** ↓ | 0.473 | 0.504 | 0.649 |
| **PRR** ↑ | 0.787 | 0.701 | 0.649 |

from 3 to 2 leads to a decrease in the Prediction Rejection Ratio ($0.787 \rightarrow 0.701$) and a slight increase in ECE. This confirms that the diversity provided by a larger set of domain experts is indeed a primary driver of high-quality uncertainty estimation. Crucially, the comparison indicates that while performance benefits from scaling up the number of domains, the method remains effective and robust when domain availability is limited.

## C  Robustness to Weak Teachers

To evaluate the robustness of DomED against weak teachers trained on limited data, we performed a stress test on the PACS dataset, specifically using Sketch as the target domain. We simulated "amateur" teachers by reducing the training data of specific source domains (Art Painting, Photo, and Cartoon) to 50% and 25%. We compared our standard strategy (equal contribution) against a down-weighted strategy, where the weak teacher's contribution to the soft target was manually suppressed (lowering the weight from 0.8 to 0.2).

As shown in Table 9, DomED proves to be remarkably resilient on the target domain. Even with only 25% of the training data, the performance drop is relatively small. Furthermore, manual down-weighting often degrades both accuracy and calibration (resulting in higher ECE). This suggests that even "amateur" teachers provide valuable diversity that aids generalization. The ensemble averaging mechanism, combined with the students hard-label loss anchor, naturally mitigates the noise from weaker teachers without requiring manual intervention.

## D  Dark Knowledge Retention

The preservation of "dark knowledge", the rich information contained in the non-target class probabilities, is essential for effective knowledge distillation (Hinton et al., 2015). However, this information is often lost when models become overly confident during training. We investigated this phenomenon by tracking the entropy of a model's predictive distribution (i.e. a categorical distribution) on the PACS dataset throughout the training process. For the student model of DomED, we track the entropy of its mean predictive distribution.

In standard knowledge distillation settings, where teachers predict on their training data, confidence typically peaks early, leading to a rapid collapse in entropy (Figure 8(a) and (b)). However, the experiment reveals a fundamentally different behavior for DomED (Figure 8(c) and (d)). Since our complementary distillation strategy requires teachers to generate predictions for domains they have never seen during training, they are effectively performing OOD inference. Consequently, they naturally yield "softer" probability distributions with stable, elevated entropy ($\approx 0.49$ nats on average) throughout the entire training process.

Table 9: Robustness of different weighting strategies to data scarcity evaluated on PACS (Target domain: Sketch).

| Data-Scarce Domain | Data % | Strategy | Student Acc (%) ↑ | ECE ↓ | NLL ↓ |
|---|---|---|---|---|---|
| *None (Baseline)* | *100%* | *Standard* | *81.6 ±0.2* | *0.028* | *0.609* |
| **Art Painting** | 50% | Standard | 81.4 ±0.3 | 0.042 | 0.597 |
| | | Down-weighted | 80.8 ±0.2 | 0.049 | 0.614 |
| | 25% | Standard | 80.5 ±0.4 | 0.050 | 0.605 |
| | | Down-weighted | 80.6 ±0.3 | 0.061 | 0.626 |
| **Photo** | 50% | Standard | 79.8 ±0.3 | 0.054 | 0.618 |
| | | Down-weighted | 80.2 ±0.2 | 0.064 | 0.605 |
| | 25% | Standard | 78.0 ±0.5 | 0.041 | 0.686 |
| | | Down-weighted | 76.5 ±0.4 | 0.069 | 0.711 |
| **Cartoon** | 50% | Standard | 77.8 ±0.3 | 0.029 | 0.671 |
| | | Down-weighted | 77.5 ±0.2 | 0.027 | 0.703 |
| | 25% | Standard | 78.6 ±0.4 | 0.041 | 0.645 |
| | | Down-weighted | 76.9 ±0.3 | 0.035 | 0.738 |

To further validate whether the retained dark knowledge is sufficient, we evaluated an intervention strategy where we doubled the distillation temperature during the late training phase. We found that this yielded no notable improvement in accuracy or calibration. This leads us to conclude that the natural "OOD signal" provided by the complementary allocation is already sufficient to preserve dark knowledge, rendering additional entropy-preserving constraints unnecessary in our framework.

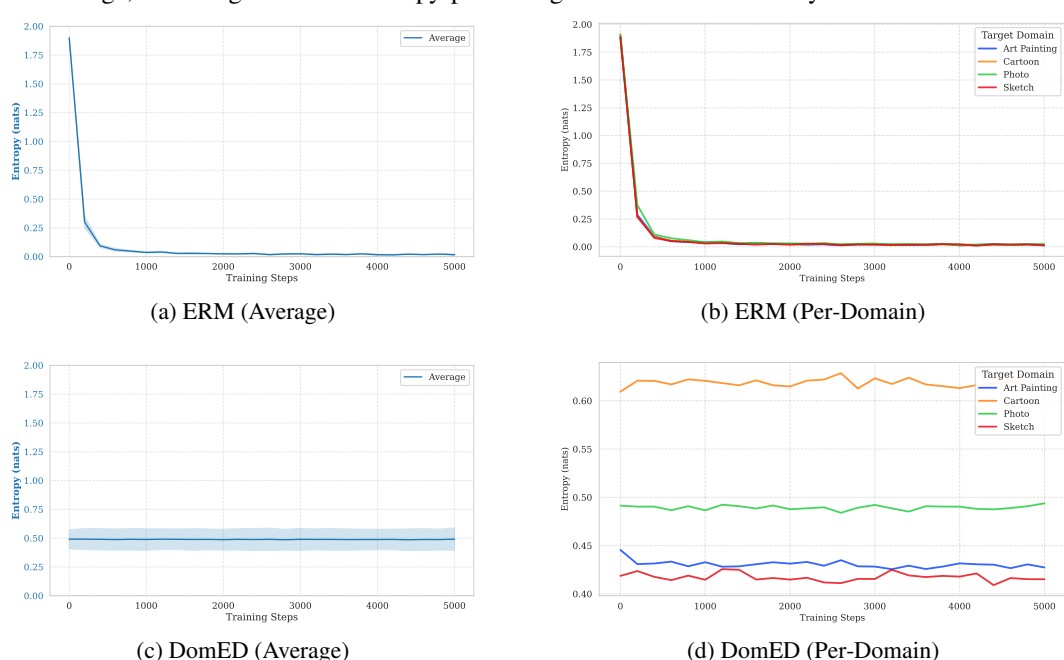

(a) ERM (Average)

(b) ERM (Per-Domain)

(c) DomED (Average)

(d) DomED (Per-Domain)

Figure 8: Entropy of the predictive distribution during model training. The average entropy (a) and per-domain entropy (b) of ERM rapidly collapse during training. In contrast, the average entropy (c) and per-domain entropy (d) of the DomED student remains stable at a high level throughout training (distillation).

# E  ABLATION ON LOSS COMPONENTS

We conduct an ablation study on the components of our proposed loss function, $\mathcal{L}'_S = \mathcal{L}_S + \beta\mathcal{L}_{\text{Dir}}$, to validate our decoupled distillation approach. This analysis compares the performance when using only the standard distillation loss ($\mathcal{L}_S$), only the Dirichlet NLL loss ($\mathcal{L}_{\text{Dir}}$), and our complete

decoupled loss. The results are presented in Table 10 (Right). Relying solely on $\mathcal{L}_{\mathrm{Dir}}$ leads to a significant degradation in classification accuracy, which is consistent with the issue of penalizing confident predictions. While $\mathcal{L}_S$ alone provides a strong performance baseline, our complete loss, $\mathcal{L}_S + \mathcal{L}_{\mathrm{Dir}}$, consistently maintains or even slightly improves accuracy across multiple benchmarks.

Table 10: Ablation study of the distillation loss components on three benchmarks, reporting classification accuracy (%).

| PACS | | | | |
| --- | --- | --- | --- | --- |
| Loss | Art P. | Cartoon | Photo | Sketch | Avg. |
| Only $\mathcal{L}_{\mathrm{Dir}}$ | 64.49 | 60.78 | 87.20 | 59.07 | 67.88 |
| Only $\mathcal{L}_S$ | 87.46 | 81.48 | 97.08 | 81.55 | 86.89 |
| $\mathcal{L}_{\mathrm{Dir}} + \mathcal{L}_S$ | **87.52** | **81.51** | **97.10** | **81.64** | **86.94** |

| VLCS | | | | |
| --- | --- | --- | --- | --- |
| Loss | Caltech | LabelMe | SUN09 | VOC07 | Avg. |
| Only $\mathcal{L}_{\mathrm{Dir}}$ | 76.15 | 57.69 | 75.36 | 69.60 | 69.70 |
| Only $\mathcal{L}_S$ | 98.26 | 65.08 | 78.23 | 78.21 | 79.94 |
| $\mathcal{L}_{\mathrm{Dir}} + \mathcal{L}_S$ | **98.51** | **65.31** | **78.42** | **78.40** | **80.14** |

| TerraIncognita | | | | |
| --- | --- | --- | --- | --- |
| Loss | Loc. 100 | Loc. 38 | Loc. 43 | Loc. 46 | Avg. |
| Only $\mathcal{L}_{\mathrm{Dir}}$ | 28.61 | 47.27 | 20.34 | 27.56 | 30.90 |
| Only $\mathcal{L}_S$ | 55.12 | 46.47 | 57.81 | **38.53** | 49.48 |
| $\mathcal{L}_{\mathrm{Dir}} + \mathcal{L}_S$ | **55.31** | **47.55** | **59.76** | 35.55 | **49.54** |

To verify whether this design effectively resolves the optimization conflict discussed in Section 2.3 (i.e., balancing mean prediction and uncertainty spread), we further evaluate accuracy and uncertainty metrics on the PACS dataset. As shown in Table 11, combining the two losses significantly improves both ECE and PRR compared to using $\mathcal{L}_{\mathrm{Dir}}$ alone. These empirical results confirm that $\mathcal{L}_S$ is essential for anchoring the mean prediction, which in turn allows $\mathcal{L}_{\mathrm{Dir}}$ to focus primarily on shaping the distribution's spread without compromising convergence.

Table 11: Impact of loss components on accuracy and uncertainty metrics evaluated on PACS.

| Loss | Acc.↑ | ECE↓ | NLL↓ | PRR↑ | K.Unc↑ |
| --- | --- | --- | --- | --- | --- |
| Only $\mathcal{L}_{\mathrm{Dir}}$ | 67.88 | 0.397 | 0.917 | 0.640 | 0.672 |
| Only $\mathcal{L}_S$ | 86.89 | 0.080 | 0.542 | 0.771 | 0.719 |
| $\mathcal{L}_{\mathrm{Dir}} + \mathcal{L}_S$ | **86.94** | **0.044** | **0.473** | **0.787** | **0.723** |

# F  SENSITIVITY TO HYPERPARAMETERS

## F.1  SOFT LABEL WEIGHT AND TEMPERATURE

We investigate the impact of the soft label weight ($\lambda$) and the temperature ($\tau$) on model accuracy. Figures 9 and 10 present heatmaps of model performance on the PACS and VLCS datasets, respectively. The analysis shows that performance varies smoothly with these hyperparameters. For instance, on the PACS dataset, the Photo domain is least sensitive, while the Cartoon domain exhibits low sensitivity to changes in $\tau$ but high sensitivity to changes in $\lambda$. Overall, these results indicate that DomED's performance is robust across a reasonable range of hyperparameter settings, validating the stability of our approach.

## F.2  WEIGHT OF THE DIRICHLET NLL LOSS

The hyperparameter $\beta$ in our decoupled distillation loss, $\mathcal{L}'_S = \mathcal{L}_S + \beta\mathcal{L}_{\mathrm{Dir}}$, is crucial for balancing accuracy and uncertainty transfer. To analyze our method's robustness to this hyperparameter, we investigate its impact on both the loss components and the final uncertainty estimates. As shown in Figure 11, we first examine the trade-off between the two loss components. Increasing $\beta$ places more weight on the uncertainty-focused $\mathcal{L}_{\mathrm{Dir}}$, causing its value to decrease at the expense of the accuracy-focused $\mathcal{L}_S$. The results indicate that a small value, such as $\beta = 0.01$, strikes an effective balance, allowing $\mathcal{L}_{\mathrm{Dir}}$ to converge sufficiently without significantly degrading the performance of $\mathcal{L}_S$.

This balance then translates directly to the final uncertainty estimates, as confirmed in Figure 12. The same region around $\beta = 0.01$ successfully preserves a high degree of both total and knowledge uncertainty from the ensemble. These results validate our core claim: the principled decoupling, controlled by a small $\beta$, effectively distills an ensemble's uncertainty without the accuracy trade-off typically associated with relying solely on the Dirichlet NLL objective.

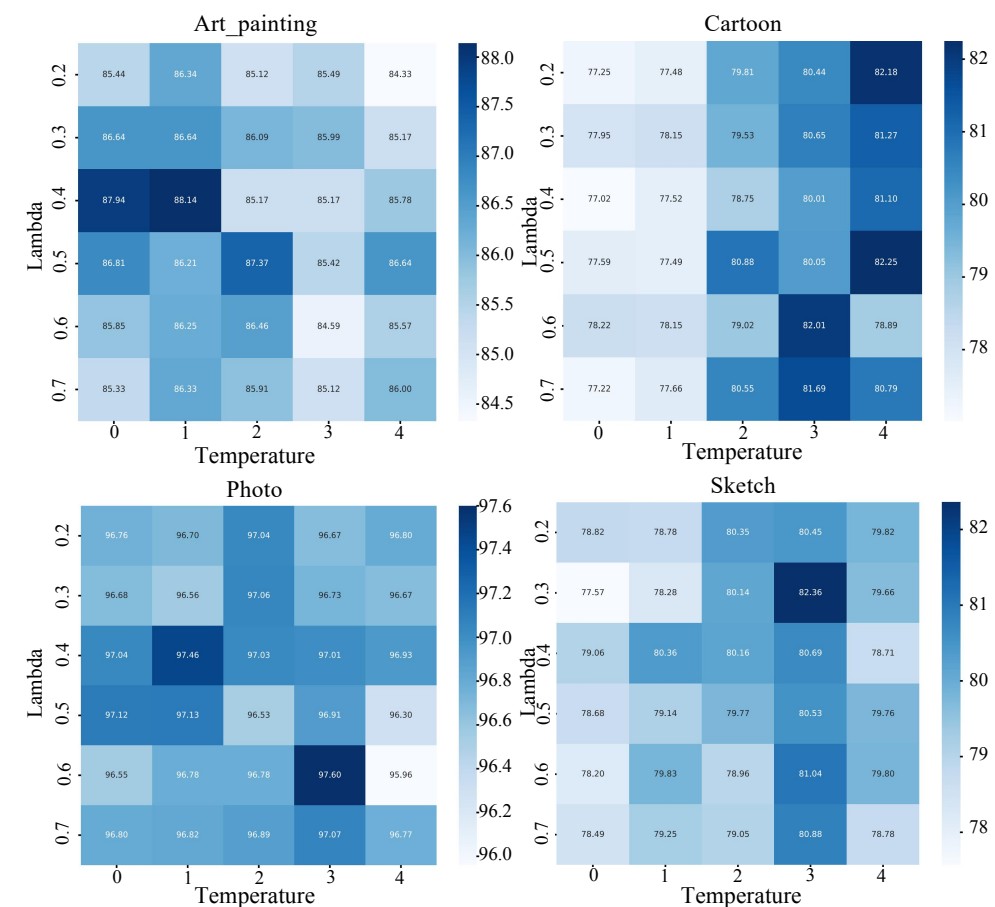

Figure 9: Test accuracies on PACS as $\lambda$ and $\tau$ vary.

## G   COMPARISON WITH ALTERNATIVE CALIBRATION STRATEGIES

We evaluated DomED against Post-hoc Temperature Scaling (TS) (Guo et al., 2017), Label Smoothing (LS) (Szegedy et al., 2016), and Focal Loss (Lin et al., 2017) on the PACS dataset. The results are summarized in Table 12.

Table 12: Comparison of DomED with alternative calibration strategies on PACS. We evaluate Post-hoc Temperature Scaling (TS), Label Smoothing (LS, $\epsilon = 0.1$), and Focal Loss ($\gamma = 2.0$) within the same complementary allocation regime.

| Category | Method / Configuration | ERR ↓ | ECE ↓ | NLL ↓ | PRR ↑ | OOD-AUC ↑ |
|---|---|---|---|---|---|---|
| Baseline | $\mathcal{L}_S$ only | 0.1311 | 0.080 | 0.542 | 0.771 | 71.96 |
| | + Post-hoc TS (Guo et al., 2017) | 0.1311 | 0.065 | 0.510 | 0.773 | 71.96 |
| Alternatives | Label Smoothing (Szegedy et al., 2016) | 0.1582 | 0.113 | 0.550 | 0.669 | **73.17** |
| | Focal Loss (Lin et al., 2017) | 0.1704 | 0.129 | 0.590 | 0.582 | 72.20 |
| Ours | **DomED** ($\mathcal{L}_S + \mathcal{L}_{Dir}$) | **0.1306** | **0.044** | **0.473** | **0.787** | 72.25 |
| | + Post-hoc TS | 0.1306 | 0.105 | 0.580 | 0.785 | 72.26 |

Post-hoc TS reduces ECE but yields negligible gains in error detection (PRR) or OOD detection. While Label Smoothing achieves a high OOD-AUC, it incurs a performance trade-off, resulting in a 2.7% increase in classification error and a significantly higher ECE compared to DomED. Focal Loss similarly results in a higher error rate and worse calibration.

In contrast, DomED achieves the best balance with the lowest error (0.1306) and ECE (0.044) while maintaining the highest PRR (0.787). Notably, applying post-hoc TS to DomED degrades calibration, suggesting that our method learns intrinsically calibrated probabilities. We attribute this strong performance partly to the fact that the Dirichlet distribution is the conjugate prior of the categorical

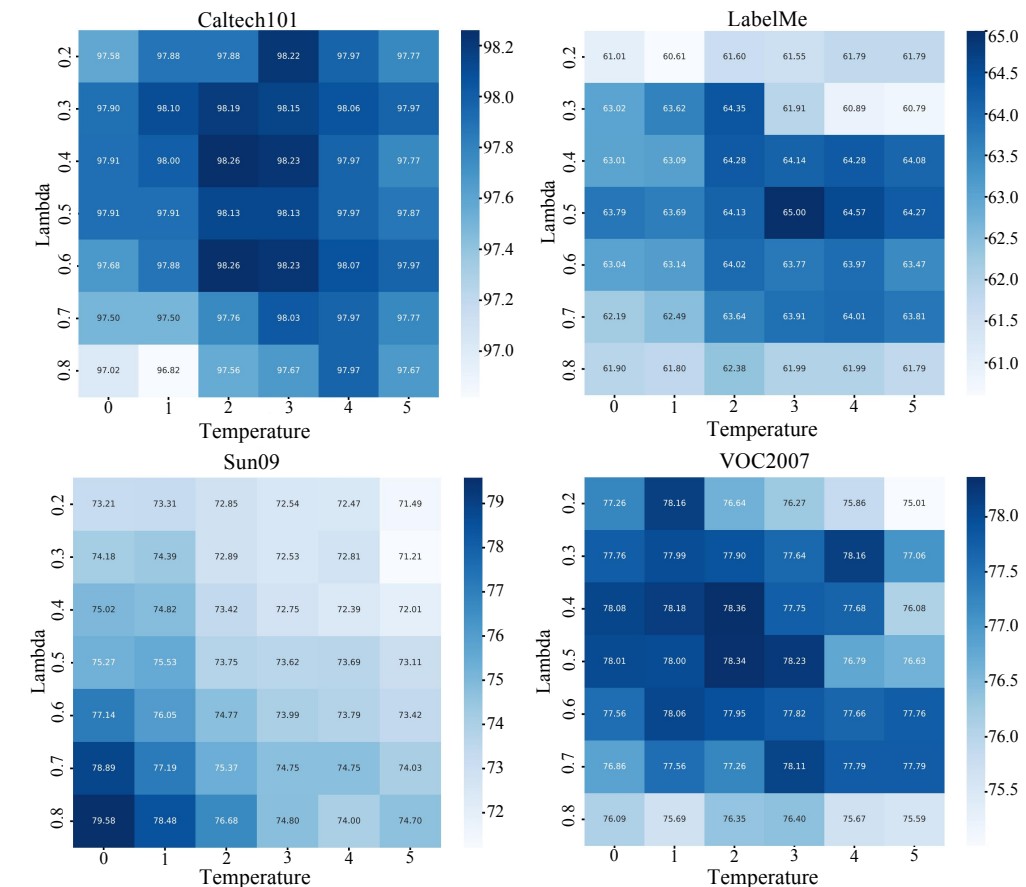

Figure 10: Test accuracies on VLCS as $\lambda$ and $\tau$ vary.

distribution. This makes it a theoretically natural choice for capturing the uncertainty inherent in an ensemble.

## H ANALYSIS OF AN ALTERNATIVE DECOUPLING STRATEGY

The standard Dirichlet NLL loss ($\mathcal{L}_{\mathrm{Dir}}$) can have a large gradient norm compared to the cross-entropy loss, which can destabilize training and harm classification accuracy. In the main paper, we address this with a simple and effective decoupled loss, $\mathcal{L}'_S = \mathcal{L}_S + \beta \mathcal{L}_{\mathrm{Dir}}$. Here, we analyze a more complex alternative to provide further justification for our chosen design. The alternative strategy aims to directly isolate the learning of the mean prediction from the uncertainty. The mean of the Dirichlet distribution is given by $\mathbb{E}[\pi_c] = \alpha_c/\alpha_0 = e^{z_c}/\sum_k e^{z_k}$, which has the same form as the standard softmax function. The uncertainty, however, is controlled by the precision $\alpha_0$, which is determined by the sum of logits, $z_0 = \sum_c z_c$. This implies that $z_0$ introduces an extra degree of freedom for learning uncertainty. Therefore, we propose decoupling the uncertainty distillation by using Eq. 7 to learn $z_0$ and Eq. 5 to learn $\mathbb{E}[\pi_c]$. This ensures that the mean categorical distribution is learned regularly without compromising model accuracy. To achieve this, we reparameterize the $z_c$'s in Eq. 7 as follows:

$$z_c = \text{stop\_gradient}(z_c - z_0) + z_0, \qquad (9)$$

where stop_gradient($\cdot$) operation blocks gradients from directly flowing to $z_c$ while allowing them to pass through $z_0$.

However, our experiments showed this approach to be highly unstable. As illustrated in Figure 13, this reparameterization causes the Dirichlet NLL to diverge for all but a very narrow range of $\beta$. When $\beta$ exceeds a small threshold (approximately 0.012), the model fails to converge, leading to a collapse in classification accuracy. This investigation highlights the difficulty of directly manipulating the gradients of the Dirichlet NLL via explicit reparameterization. It motivated our decision to use the simpler and far more robust decoupled loss function presented in the main paper, which

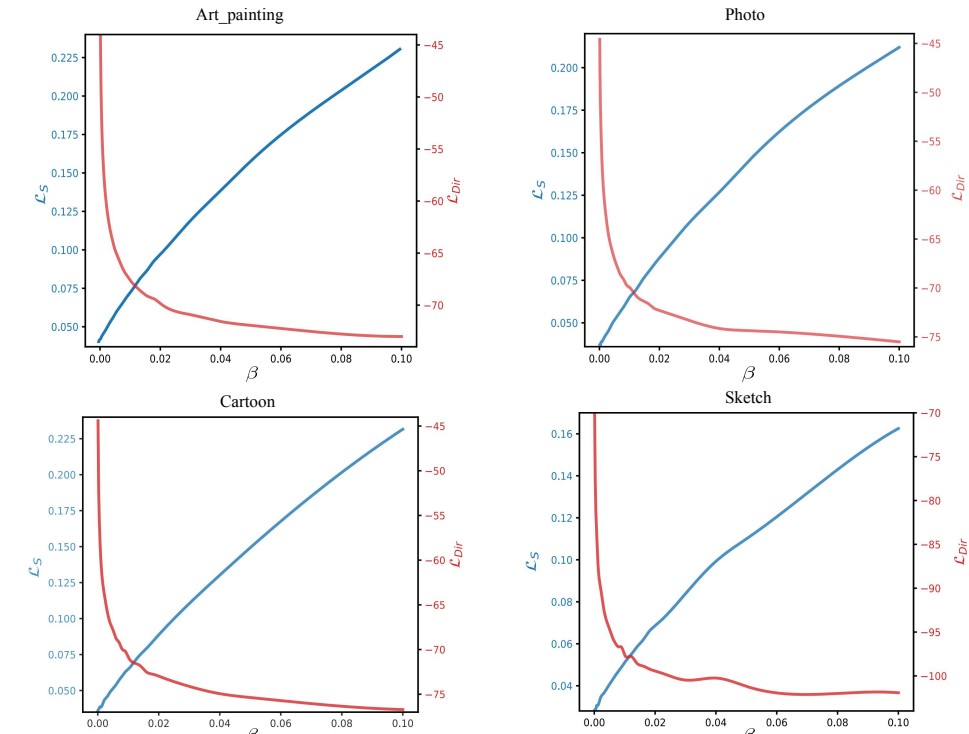

Figure 11: Converged values of the loss components $\mathcal{L}_S$ (blue) and $\mathcal{L}_{\text{Dir}}$ (red) as a function of $\beta$. The plot illustrates the trade-off between optimizing for accuracy ($\mathcal{L}_S$) and uncertainty ($\mathcal{L}_{\text{Dir}}$) on the PACS dataset.

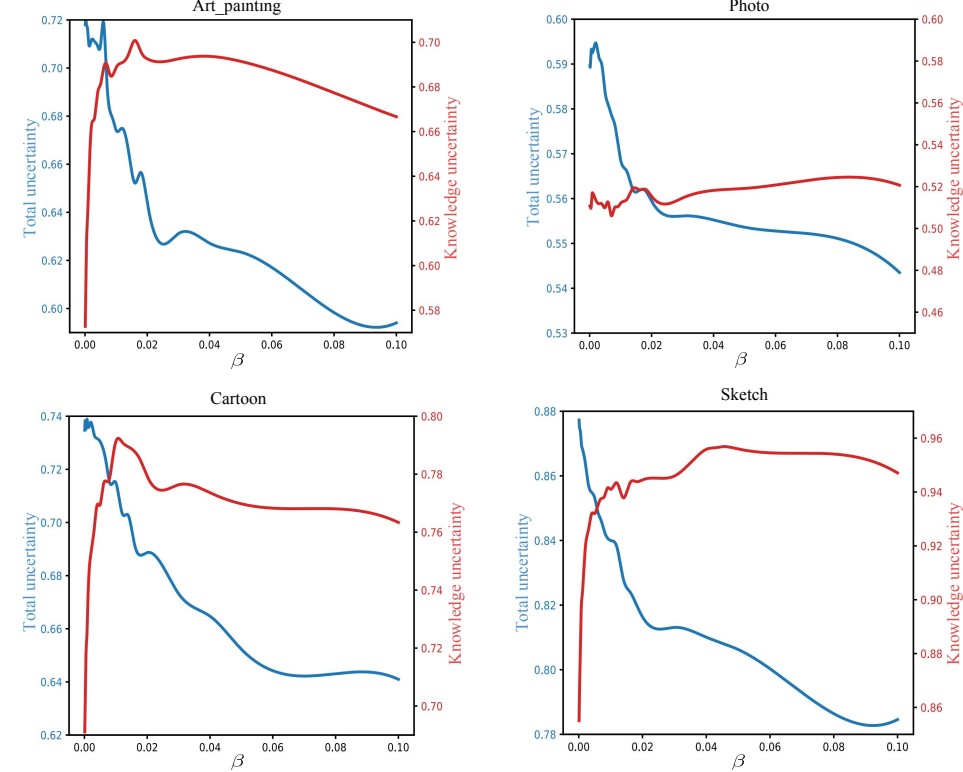

Figure 12: Impact of $\beta$ on the final uncertainty estimates for the PACS dataset. The plots show that a small $\beta$ (e.g., 0.01) is sufficient to distill a high degree of both total and knowledge uncertainty from the teacher ensemble.

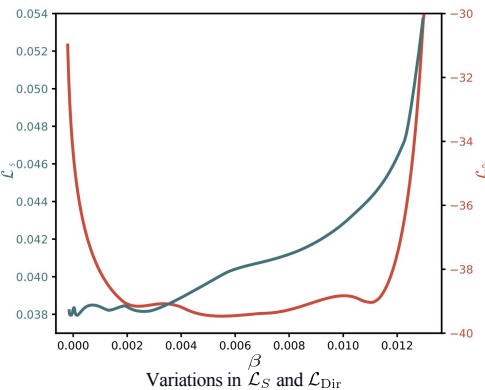
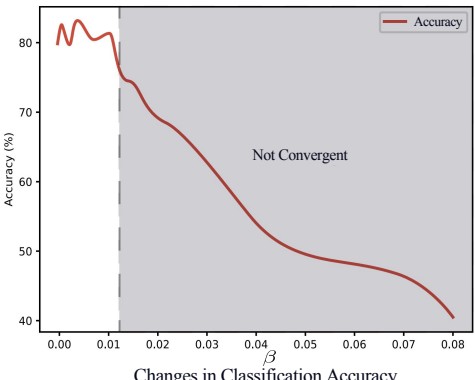

Figure 13: The alternative decoupling strategy using 'stop_gradient' is highly unstable. (Left) Both loss components, $\mathcal{L}_S$ and $\mathcal{L}_{\text{Dir}}$, diverge with increasing $\beta$ on the PACS dataset. (Right) Classification accuracy collapses when $\beta$ exceeds a small threshold, indicating a failure to converge.

effectively balances the two objectives without requiring explicit gradient surgery and demonstrates stable performance across all benchmarks.

## I ROBUSTNESS TO MODEL ARCHITECTURE

We evaluate the architectural robustness of DomED by repeating our experiments on the PACS dataset with a smaller CNN, ResNet-18 (He et al., 2016), and a Vision Transformer (ViT-B/16) (Dosovitskiy et al., 2021). As shown in Table 13, the results confirm that DomED is a broadly applicable framework that consistently improves upon the ERM baseline. With the smaller ResNet-18, DomED not only improves accuracy but also yields superior uncertainty estimates (ECE and NLL) compared to the full test-time ensemble, demonstrating its effectiveness on more compact models. The results on the ViT-B/16 are particularly compelling. Vision Transformers are known to struggle with generalization on smaller DG datasets, which is reflected in the high error rates of the ERM and ensemble baselines. In this challenging scenario,

Table 13: Performance on PACS across diverse model architectures (Arch.), including ResNet-18 (R-18) and the Vision Transformer Base model (ViT-B/16). DomED consistently improves accuracy and uncertainty estimation over the ERM baselines, demonstrating its robustness.

| Arch. | Model | ERR↓ | ECE↓ | NLL↓ | PRR↑ | T.Unc | K.Unc |
|---|---|---|---|---|---|---|---|
| R-18 | ERM | 0.192 | 0.128 | 0.895 | **0.775** | 0.723 | - |
| | Ensemble | **0.172** | 0.066 | 0.697 | 0.772 | **0.724** | 0.698 |
| | DomED | 0.175 | **0.043** | **0.579** | 0.745 | 0.713 | **0.729** |
| R-50 | ERM | 0.149 | 0.089 | 0.625 | 0.776 | 0.714 | - |
| | Ensemble | **0.130** | **0.038** | **0.471** | **0.798** | **0.730** | 0.711 |
| | DomED | 0.131 | 0.044 | 0.473 | 0.787 | 0.705 | **0.723** |
| ViT-B/16 | ERM | 0.211 | 0.155 | 0.988 | 0.688 | 0.723 | - |
| | Ensemble | 0.203 | **0.050** | 0.722 | 0.711 | **0.724** | **0.721** |
| | DomED | **0.146** | 0.079 | **0.534** | **0.785** | 0.623 | 0.692 |

DomED provides a remarkable improvement, reducing the classification error by approximately 6 percentage points compared to the ensemble. This suggests that our distillation strategy may function as a powerful regularizer, validating DomED as a robust framework for diverse model families in the domain generalization setting.

## J ADDITIONAL RESULTS ON UNCERTAINTY QUANTIFICATION

This section provides a detailed breakdown of the uncertainty quantification results summarized in Section 3.3. We present the performance metrics for each leave-one-domain-out split, supplementing the averaged results shown in the main paper. Table 14 contains the per-domain calibration and reliability metrics corresponding to Table 3. Table 15 provides the per-domain OOD detection performance corresponding to Table 4.

Table 14: Detailed uncertainty quantification results on the PACS dataset. The table is split into two parts for better readability: Art Painting and Cartoon (top), Photo and Sketch (bottom). The best and second-best results are indicated by bold and underlined, respectively.

| | *Target: Art Painting* | | | | *Target: Cartoon* | | | |
|---|---|---|---|---|---|---|---|---|
| **Model** | **ERR↓** | **ECE↓** | **NLL↓** | **PRR↑** | **ERR↓** | **ECE↓** | **NLL↓** | **PRR↑** |
| Ensemble | 0.139 | 0.026 | 0.489 | 0.830 | 0.160 | 0.057 | 0.633 | 0.761 |
| EoA | 0.111 | 0.048 | 0.375 | 0.894 | 0.143 | 0.099 | 0.627 | 0.774 |
| ERM | 0.152 | 0.065 | 0.496 | **0.808** | 0.200 | 0.125 | 0.821 | 0.731 |
| Temp. Scaling | 0.164 | 0.069 | 0.576 | 0.759 | 0.191 | 0.061 | **0.599** | **0.758** |
| MC Drop (p=0.5) | 0.163 | 0.100 | 0.686 | 0.772 | 0.238 | 0.189 | 1.414 | 0.678 |
| MC Drop (p=0.1) | 0.175 | 0.109 | 0.736 | 0.766 | 0.203 | 0.165 | 1.241 | 0.703 |
| CORAL | 0.139 | 0.071 | 0.514 | 0.670 | 0.181 | 0.117 | 0.754 | 0.484 |
| EnD$^2$ | 0.274 | 0.131 | 1.262 | 0.142 | 0.481 | 0.358 | 2.590 | 0.007 |
| DomED Teachers | 0.269 | 0.080 | 0.847 | 0.657 | 0.401 | 0.074 | 1.456 | 0.386 |
| Scheme (c) | 0.128 | 0.060 | 0.490 | 0.795 | 0.190 | 0.042 | 0.698 | 0.679 |
| Scheme (d) | **0.125** | 0.125 | **0.477** | 0.748 | **0.179** | 0.053 | 0.697 | 0.710 |
| DomED (Ours) | **0.125** | 0.049 | 0.497 | 0.781 | 0.185 | **0.038** | 0.628 | 0.738 |

| | *Target: Photo* | | | | *Target: Sketch* | | | |
|---|---|---|---|---|---|---|---|---|
| **Model** | **ERR↓** | **ECE↓** | **NLL↓** | **PRR↑** | **ERR↓** | **ECE↓** | **NLL↓** | **PRR↑** |
| Ensemble | 0.038 | 0.012 | 0.125 | 0.945 | 0.184 | 0.057 | 0.637 | 0.657 |
| EoA | 0.019 | 0.011 | 0.049 | 0.958 | 0.166 | 0.076 | 0.561 | 0.722 |
| ERM | 0.038 | 0.045 | 0.265 | 0.891 | 0.207 | 0.121 | 0.919 | 0.674 |
| Temp. Scaling | 0.037 | **0.018** | **0.147** | 0.902 | 0.197 | 0.045 | 0.626 | 0.672 |
| MC Drop (p=0.5) | 0.049 | 0.032 | 0.219 | 0.904 | 0.250 | 0.196 | 1.173 | 0.631 |
| MC Drop (p=0.1) | 0.059 | 0.044 | 0.290 | 0.881 | 0.191 | 0.131 | 0.754 | **0.726** |
| CORAL | 0.032 | 0.026 | 0.158 | 0.887 | 0.201 | 0.103 | 0.720 | 0.364 |
| EnD$^2$ | 0.145 | 0.124 | 0.696 | 0.125 | 0.405 | 0.269 | 1.773 | 0.060 |
| DomED Teachers | 0.194 | 0.200 | 0.943 | 0.482 | 0.261 | 0.237 | 1.006 | 0.620 |
| Scheme (c) | 0.030 | 0.077 | 0.237 | 0.895 | 0.186 | 0.038 | **0.583** | 0.635 |
| Scheme (d) | 0.035 | 0.096 | 0.229 | 0.890 | 0.194 | 0.090 | 0.690 | 0.625 |
| DomED (Ours) | **0.029** | 0.061 | 0.158 | **0.936** | 0.184 | **0.028** | 0.609 | 0.694 |

Table 15: Out-of-distribution detection performance across four datasets, evaluated using the ROC-AUC metric. For each dataset, OOD samples are drawn from the test domain, while in-distribution (ID) samples are drawn from the train domains.

| | *PACS* | | | | | | | | | |
|---|---|---|---|---|---|---|---|---|---|---|
| | Art Painting | | Cartoon | | Photo | | Sketch | | Avg. | |
| **Model** | **T.Unc** | **K.Unc** | **T.Unc** | **K.Unc** | **T.Unc** | **K.Unc** | **T.Unc** | **K.Unc** | **T.Unc** | **K.Unc** |
| ERM | 0.73 | – | 0.69 | – | 0.57 | – | 0.89 | – | 0.71 | – |
| Ensemble | 0.75 | 0.71 | 0.73 | 0.76 | 0.54 | 0.46 | 0.90 | 0.91 | 0.73 | 0.71 |
| DomED | 0.69 | 0.72 | 0.72 | 0.77 | 0.57 | 0.47 | 0.84 | 0.93 | 0.71 | 0.72 |

| | *OfficeHome* | | | | | | | | | |
|---|---|---|---|---|---|---|---|---|---|---|
| | Art | | Clipart | | Product | | Real World | | Avg. | |
| **Model** | **T.Unc** | **K.Unc** | **T.Unc** | **K.Unc** | **T.Unc** | **K.Unc** | **T.Unc** | **K.Unc** | **T.Unc** | **K.Unc** |
| ERM | 0.72 | – | 0.79 | – | 0.62 | – | 0.57 | – | 0.67 | – |
| Ensemble | 0.75 | 0.65 | 0.81 | 0.84 | 0.63 | 0.68 | 0.58 | 0.65 | 0.69 | 0.70 |
| DomED | 0.69 | 0.71 | 0.72 | 0.79 | 0.51 | 0.66 | 0.50 | 0.64 | 0.60 | 0.70 |

| | *VLCS* | | | | | | | | | |
|---|---|---|---|---|---|---|---|---|---|---|
| | Caltech101 | | LabelMe | | SUN09 | | VOC2007 | | Avg. | |
| **Model** | **T.Unc** | **K.Unc** | **T.Unc** | **K.Unc** | **T.Unc** | **K.Unc** | **T.Unc** | **K.Unc** | **T.Unc** | **K.Unc** |
| ERM | 0.32 | – | 0.43 | – | 0.66 | – | 0.65 | – | 0.51 | – |
| Ensemble | 0.33 | 0.23 | 0.45 | 0.51 | 0.67 | 0.72 | 0.66 | 0.71 | 0.53 | 0.54 |
| DomED | 0.48 | 0.51 | 0.44 | 0.58 | 0.69 | 0.83 | 0.67 | 0.85 | 0.57 | 0.69 |

| | *TerraIncognita* | | | | | | | | | |
|---|---|---|---|---|---|---|---|---|---|---|
| | Location 100 | | Location 38 | | Location 43 | | Location 46 | | Avg. | |
| **Model** | **T.Unc** | **K.Unc** | **T.Unc** | **K.Unc** | **T.Unc** | **K.Unc** | **T.Unc** | **K.Unc** | **T.Unc** | **K.Unc** |
| ERM | 0.74 | – | 0.70 | – | 0.76 | – | 0.79 | – | 0.75 | – |
| Ensemble | 0.79 | 0.80 | 0.76 | 0.87 | 0.74 | 0.60 | 0.85 | 0.90 | 0.78 | 0.79 |
| DomED | 0.75 | 0.66 | 0.74 | 0.86 | 0.74 | 0.60 | 0.83 | 0.87 | 0.77 | 0.75 |

## K   ETHICS STATEMENT

This work focuses on foundational machine learning for domain generalization. All experiments were conducted on publicly available and widely used academic datasets (e.g., PACS, OfficeHome), which, to our knowledge, do not contain personally identifiable or sensitive content. Our research did not involve human subjects, and we do not foresee any direct negative societal impacts stemming from our methodology or findings.

## L   REPRODUCIBILITY STATEMENT

Our full implementation, built upon PyTorch and the DomainBed framework, is provided in the supplementary material. All experiments were conducted on NVIDIA V100 GPUs, following the standard data splits and evaluation protocols established by DomainBed. Key hyperparameters were selected using training-domain validation. A comprehensive list of hyperparameters and example execution commands are available in the supplementary material's README file.

## M   USE OF LARGE LANGUAGE MODELS

We used Google's Gemini 2.5 Pro (Team et al., 2023) to correct grammatical errors and refine the language in this manuscript. The model's role was strictly that of a proofreading tool to improve clarity. All core scientific contributions, including research ideation, methodological development, experimental design, and the interpretation of results, were performed exclusively by the human authors.

