# OpenReview forum: "DomED: Redesigning Ensemble Distillation for Domain Generalization"
_ICLR.cc/2026/Conference — Submitted to ICLR 2026_

### Official Review · Reviewer_6UZY · 2025-10-29

**Soundness:** 3
**Presentation:** 3
**Contribution:** 3
**Rating:** 6
**Confidence:** 4

**Summary:**

The paper proposes DomED, a domain-aware ensemble distillation framework for domain generalization. Two ideas are central. First, complementary domain allocation trains each teacher on a single source domain and distills the student on complementary domains, which is compared against five alternative allocation schemes with consistent advantages in accuracy and calibration. Second, the loss decouples accuracy and uncertainty by combining a standard distillation objective with a Dirichlet NLL regularizer using a small weight. The method is evaluated on PACS, OfficeHome, VLCS, TerraIncognita, and DomainNet using the DomainBed protocol. Results show stable accuracy gains over ERM and clear calibration benefits, with single model inference that approaches test-time ensembles on several reliability metrics.

**Strengths:**

- **Design aligned with DG structure.** Complementary domain allocation is systematically compared against multiple alternatives and yields small but consistent accuracy gains together with stronger calibration.

- **Uncertainty-preserving distillation.** The decoupled objective $L_S + beta * L_Dir$ avoids the accuracy degradation observed with pure Dirichlet training and improves *ECE* and *NLL*.

- **Pragmatic efficiency observation.** Student performance stabilizes early, which motivates training each teacher for only a fraction of the nominal steps and supports a practical efficiency narrative.

**Weaknesses:**

- **Lack of student-side quantification for teacher count and domain granularity**
Appendix B shows that single-domain teachers exhibit higher diversity than dual-domain teachers. However, the manuscript does not report how this diversity translates to the student when the number of teachers *M* or the granularity of domain splits changes.

- **TerraIncognita requires a dataset-specific analysis**
In the DomainBed summary, the improvement on TerraIncognita is modest relative to ERM and notably smaller than on OfficeHome. In addition, on TerraIncognita the ROC-AUC for DomED is slightly below that of a three-model ensemble. The paper provides a mechanism-level explanation for OfficeHome but presents only aggregate figures for TerraIncognita. A brief, dataset-specific analysis for TerraIncognita would complete the narrative.

- **Related work on parameter-efficient fusion is under-covered**
The manuscript situates DomED relative to weight averaging and ensembling but does not cover parameter-efficient expert aggregation that is conceptually adjacent to DomED’s goal of integrating multiple experts. A concise subsection contrasting DomED with representative lines such as AdapterFusion and Mixture-of-Adapters, LoRA-based fusion and merging strategies, and prompt or prefix compositions would sharpen the novelty boundary.

- **Wording relative to test-time ensembles should strictly match Table 3**
Table 3 indicates that on PACS the EoA ensemble outperforms DomED on ERR, NLL, and PRR, while DomED achieves a lower ECE. Hence, the text should not suggest that DomED broadly approaches ensemble performance.

**Questions:**

- **Ablation on dark knowledge retention**
As teachers become sharper during late training, non-target probabilities shrink and dark knowledge diminishes. Did you evaluate higher distillation temperatures or entropy-preserving constraints in the late phase to retain dark knowledge, and does this materially improve DomED’s accuracy or calibration under complementary allocation

- **Why Dirichlet NLL instead of simpler or alternative proper scoring rules**
Your calibration gains are attributed to the Dirichlet regularizer. Can the same gains be achieved with strong but simpler single-model calibrators such as temperature scaling, label smoothing with tuned schedules, or focal-style cross entropy, when used in the same complementary allocation regime Further, did you compare against other strictly proper scoring rules for probabilistic classification such as Brier score with class-wise weighting or energy-based calibration penalties

---

> ### Author Response · Authors · 2025-11-27
> **Response to Reviewer 6UZY (1/3)**
>
> We sincerely thank the reviewer for the constructive feedback and for acknowledging our design's alignment with DG structures and the pragmatic efficiency of DomED. We address your specific concerns and questions below.
>
> **Response to W1: Lack of Student-side Quantification for Teacher Count and Domain Granularity**
>
> We appreciate the suggestion to quantify how teacher diversity translates to student performance. We have added two specific analyses to Appendix B.2 and Appendix B.3 to address this.
>
> We compared our proposed Single-domain allocation (high granularity, high diversity) against a Dual-domain allocation (low granularity, low diversity) on the PACS dataset. As shown below, the higher diversity provided by single-domain teachers translates into superior student performance. While accuracy improves by 2.0%, the impact on uncertainty is most notable: the single-domain strategy reduces Expected Calibration Error (ECE) by over 50% (0.107 $\to$ 0.044).
>
> **Table R1: Impact of teacher diversity: Comparison between single-domain (DomED) and dual-domain allocation strategies on PACS.**
>
> | Target Domain | Allocation | Diversity | Acc. (%) $\uparrow$ | ECE $\downarrow$ | NLL $\downarrow$ | PRR $\uparrow$ |
> | :--- | :--- | :--- | :---: | :---: | :---: | :---: |
> | **Art Painting** | Dual-domain | Low | 85.1 | 0.109 | 0.521 | 0.772 |
> | | **Single-domain (DomED)** | **High** | **87.5** | **0.049** | **0.497** | **0.781** |
> | **Cartoon** | Dual-domain | Low | 78.3 | 0.076 | 0.726 | 0.654 |
> | | **Single-domain (DomED)** | **High** | **81.5** | **0.038** | **0.628** | **0.738** |
> | **Photo** | Dual-domain | Low | 96.0 | 0.094 | 0.200 | 0.882 |
> | | **Single-domain (DomED)** | **High** | **97.1** | **0.061** | **0.158** | **0.936** |
> | **Sketch** | Dual-domain | Low | 80.2 | 0.150 | 0.716 | 0.614 |
> | | **Single-domain (DomED)** | **High** | **81.6** | **0.028** | **0.609** | **0.694** |
> | **Average** | Dual-domain | Low | 84.9 | 0.107 | 0.541 | 0.730 |
> | | **Single-domain (DomED)** | **High** | **86.9** | **0.044** | **0.473** | **0.787** |
>
> To quantify the impact of ensemble size (teacher count), we conducted an ablation study on the PACS dataset. We systematically train models on all possible 2-domain subsets ($M=2$, each resulting in 2 domain-specific teachers) and compare their average performance against the standard 3-domain configuration ($M=3$) as well as an ERM baseline trained on the same 2-domain subsets. The results are summarized below (and detailed in Appendix B.3):
>
> **Table R2: Impact of ensemble size ($M$) on student performance evaluated on PACS.**
>
> | Metric | DomED ($M=3$) | DomED ($M=2$) | ERM ($M=2$) |
> | :--- | :---: | :---: | :---: |
> | Accuracy $\uparrow$ | 86.9% $\pm$0.2 | 84.6% $\pm$0.3 | 84.3% $\pm$0.2 |
> | ECE $\downarrow$ | 0.044 | 0.050 | 0.097 |
> | NLL $\downarrow$ | 0.473 | 0.504 | 0.649 |
> | PRR $\uparrow$ | 0.787 | 0.701 | 0.649 |
>
> These results lead to two key observations: 1) Despite the reduction in teachers, DomED ($M=2$) still outperforms the ERM baseline trained on the identical data subset. While accuracy is comparable, DomED maintains significantly better calibration (ECE 0.050 vs. 0.097). 2) Reducing the ensemble size from 3 to 2 leads to a decrease in the Prediction Rejection Ratio (0.787 $\to$ 0.701) and a slight increase in ECE. This confirms that the diversity provided by a larger set of domain experts is indeed a primary driver of high-quality uncertainty estimation.
>
> This indicates that while performance benefits from scaling up the number of domains, the method remains effective and robust when domain availability is limited.
>
> **Response to W2: TerraIncognita Analysis and ROC-AUC Gap**
>
> We have added a dataset-specific analysis for TerraIncognita in Appendix B.1. As visualized in the new accuracy heatmaps, single-domain teachers on TerraIncognita exhibit a pattern of high specialization, achieving high accuracy on their source domains but suffering significant performance drops on unseen domains. We hypothesize that this behavior may stem from the nature of camera trap data, where static backgrounds (e.g., specific vegetation or terrain) can encourage models to learn domain-specific shortcuts that do not transfer well.
>
> Despite this challenging scenario where teacher transferability is limited, DomED effectively recovers useful signals to outperform the ERM baseline (50.2% vs 47.8%). Regarding the slight gap in ROC-AUC compared to the full ensemble, we suggest this is a consequence of distillation compression. The teacher ensemble likely exhibits high variance in its predictions on this dataset (swinging between high confidence and errors). Compressing this complex uncertainty landscape into a single static Dirichlet prior is inherently challenging, leading to a minor loss in OOD detection capability compared to the full ensemble.

---

> ### Author Response · Authors · 2025-11-27
> **Response to Reviewer 6UZY (2/3)**
>
> **Response to W3: Related Work on Parameter-Efficient Fusion**
>
> We appreciate the reviewer's suggestion to position DomED relative to parameter-efficient expert aggregation. We have expanded Section 4.2 to discuss methods such as AdapterFusion [1] and LoRA-based merging [2, 3]. DomED distinguishes itself from these approaches in two fundamental ways. First, while parameter-efficient methods typically operate in weight or feature space to route or merge experts, DomED operates in the output space via distillation, allowing it to be model-agnostic regarding the architecture of the teachers. Second, whereas most fusion methods focus primarily on maximizing predictive accuracy, DomED is explicitly designed to aggregate and transfer epistemic uncertainty through our decoupled Dirichlet objective, addressing a different dimension of model reliability. Finally, we clarify that the weight averaging (SWAD) employed in our experiments was included to demonstrate the compatibility of our distillation framework with standard optimization techniques in the DG community [4, 5], rather than to position parameter merging as a core contribution of our work.
>
> **Response to W4: Wording Relative to Test-Time Ensembles**
>
> We agree with the reviewer’s observation and have revised the manuscript to reflect this distinction accurately. We have updated the text and the caption of Table 3 to clarify that while DomED achieves the best calibration (ECE) among all methods, the computationally expensive test-time ensembles (like EoA) still provide a performance ceiling on metrics like ERR and NLL. The text now states: "Overall, DomED achieves the best performance among all single-inference methods ... transferring the benefits of an ensemble into a single efficient model." Similarly, the caption for Table 3 has been updated to remove any suggestion of broad equivalence.
>
> **Response to Q1: Ablation on Dark Knowledge Retention**
>
> We thank the reviewer for this insightful question regarding the potential loss of dark knowledge. We investigated this phenomenon by tracking the entropy of a model's predictive distribution (i.e. a categorical distribution) on the PACS dataset throughout training.
>
> In standard knowledge distillation settings, where teachers predict on their training data, confidence typically peaks early, leading to a rapid collapse in entropy. However, our analysis (see Figure 8 in Appendix D) reveals a fundamentally different behavior for DomED. Because our complementary distillation" strategy requires teachers to generate predictions for domains they have never seen during training, they are effectively performing OOD inference. Consequently, they naturally yield "softer" probability distributions with stable, elevated entropy ($\approx 0.49$ nats on average) throughout the entire training process.
>
> To directly address your question, we evaluated an intervention strategy where we doubled the distillation temperature during the late training phase. We found that this yielded no notable improvement in accuracy or calibration. This leads us to conclude that the natural "OOD signal" provided by the complementary allocation is already sufficient to preserve dark knowledge, rendering entropy-preserving constraints unnecessary in our framework.

---

> ### Author Response · Authors · 2025-11-27
> **Response to Reviewer 6UZY (3/3)**
>
> **Response to Q2: Comparison with Alternative Calibration Strategies**
>
> We evaluated DomED against Post-hoc Temperature Scaling (TS) [6], Label Smoothing (LS) [7], and Focal Loss [8] on the PACS dataset. The results are summarized in the table below (and included in Appendix G).
>
> Post-hoc TS reduces ECE but yields negligible gains in error detection (PRR) or OOD detection. While Label Smoothing achieves a high OOD-AUC, it incurs a performance trade-off, resulting in a 2.7% increase in classification error and a significantly higher ECE compared to DomED. Focal Loss similarly results in a higher error rate and worse calibration. In contrast, DomED achieves the best balance with the lowest error (0.1306) and ECE (0.044) while maintaining the highest PRR (0.787). Notably, applying post-hoc TS to DomED degrades calibration, suggesting that our method learns intrinsically calibrated probabilities.
>
> We attribute this strong performance partly to the fact that the Dirichlet distribution is the conjugate prior of the categorical distribution. This makes it a theoretically natural choice for capturing the uncertainty inherent in an ensemble and proves empirically effective in our setting. We sincerely thank the reviewer for suggesting alternative scoring rules such as the Brier score or energy-based calibration penalties, and agree that exploring these principled alternatives is a valuable direction for future research.
>
> **Table R2: Comparison of DomED with alternative calibration strategies on PACS.**
>
> | Category | Method / Configuration | ERR $\downarrow$ | ECE $\downarrow$ | NLL $\downarrow$ | PRR $\uparrow$ | OOD-AUC $\uparrow$ |
> | :--- | :--- | :---: | :---: | :---: | :---: | :---: |
> | **Baseline** | $\mathcal{L}\_S$ only | 0.1311 | 0.080 | 0.542 | 0.771 | 71.96 |
> | | + Post-hoc TS [6] | 0.1311 | 0.065 | 0.510 | 0.773 | 71.96 |
> | **Alternatives** | Label Smoothing [7] | 0.1582 | 0.113 | 0.550 | 0.669 | **73.17** |
> | | Focal Loss [8] | 0.1704 | 0.129 | 0.590 | 0.582 | 72.20 |
> | **Ours** | **DomED ($\mathcal{L}\_S + \mathcal{L}\_{Dir}$)** | **0.1306** | **0.044** | **0.473** | **0.787** | 72.25 |
> | | + Post-hoc TS | 0.1306 | 0.105 | 0.580 | 0.785 | 72.26 |
>
> References:
>
> [1] J. Pfeiffer et al., "Adapterfusion: Non-destructive task composition for transfer learning," in EACL, 2021.
>
> [2] E. J. Hu et al., "Lora: Low-rank adaptation of large language models," in ICLR, 2022.
>
> [3] P. Yadav et al., "Ties-merging: Resolving interference when merging models," in NeurIPS, 2023.
>
> [4] J. Cha et al., "Swad: Domain generalization by seeking flat minima," in NeurIPS, 2021.
>
> [5] X. Chu et al., "DNA: Domain generalization with diversified neural averaging," in ICML, 2022.
>
> [6] C. Guo, G. Pleiss, Y. Sun, and K. Q. Weinberger, "On calibration of modern neural networks," in ICML, 2017.
>
> [7] C. Szegedy et al., "Rethinking the inception architecture for computer vision," in CVPR, 2016.
>
> [8] T. Lin et al., "Focal loss for dense object detection," in ICCV, 2017.

---

### Official Review · Reviewer_XJwQ · 2025-10-30

**Soundness:** 3
**Presentation:** 3
**Contribution:** 2
**Rating:** 4
**Confidence:** 5

**Summary:**

This paper proposes DomED (Domain-aware Ensemble Distillation), a tailored ensemble–distillation framework for domain generalization (DG) that (i) trains domain-specific teachers, each on a single source domain to promote diversity; (ii) performs complementary distillation, where teachers distill only on unseen (complementary) source domains to encourage cross-domain generalization; and (iii) uses decoupled uncertainty-preserving distillation, separating accuracy and uncertainty transfer via a cross-entropy loss and a Dirichlet-prior loss, respectively, to address the trade-off between calibration and accuracy.

**Strengths:**

1. Clear motivation and formulation. The paper clearly identifies the overlooked role of uncertainty estimation in DG and provides a principled way to integrate ensemble distillation with uncertainty transfer.
2. Novel decoupled uncertainty distillation. The split between accuracy-driven and uncertainty-driven loss terms is conceptually clean and empirically validated.
3. Comprehensive evaluation. The experiments span multiple DG benchmarks and cover both accuracy and uncertainty metrics.

**Weaknesses:**

1. Limited novelty beyond recombination of known components. The ideas of domain-specific ensembling and uncertainty-aware distillation are incremental extensions of existing work [1]
2. Insufficient theoretical analysis. The method is motivated intuitively but lacks theoretical grounding for why complementary data allocation improves generalization or why the decoupling of uncertainty loss preserves accuracy.
3. Scalability concerns. Despite claiming efficiency, training multiple domain-specific teachers (especially on DomainNet) can be computationally heavy. The paper does not report compute/efficiency metrics for training M domain-specific teachers plus the student, nor cost-normalized accuracy/calibration comparisons against single-model or weight-averaged baselines; scalability with M remains unclear
4. Modest accuracy gains. Improvements over strong baselines.
5. he evaluation under-represents recent knowledge-distillation–based DG methods [2][3][4]

[1] 2022 - ECCV - Cross-domain ensemble distillation for domain generalization

[2] 2021 - ACM MM - Embracing the Dark Knowledge: Domain Generalization Using Regularized Knowledge Distillation

[3] 2023 - ICCV - A Sentence Speaks a Thousand Images: Domain Generalization through Distilling CLIP with Language Guidance

[4] 2025 - IJCAI - Balancing Invariant and Specific Knowledge for Domain Generalization with Online Knowledge Distillation

**Questions:**

See Weakness.

---

> ### Author Response · Authors · 2025-11-27
> **Response to Reviewer XjwQ (1/3)**
>
> We sincerely thank Reviewer XjwQ for the detailed assessment and for recognizing the clear motivation, clean formulation of our decoupled objective, and the comprehensive evaluation. We address your concerns regarding novelty, theory, and scalability below.
>
> **Response to W1: Novelty and Comparison with XDED**
>
> We appreciate the opportunity to clarify the distinction between DomED and XDED [1]. While both involve distillation, they use fundamentally different mechanisms. XDED is a self-distillation method that distills an ensemble of the output logits (of the same model) from training data with the same label, *ignoring the domain labels*. In contrast, DomED constructs an explicit ensemble of domain-specialized teachers by using the domain labels. This difference is crucial for both generalization and calibration, since a single model (XDED) cannot capture the "disagreement" between diverse experts.
>
> For empirical comparison, we note that the absence of publicly available code for XDED makes a direct side-by-side comparison difficult. Nevertheless, we take their reported results on DomainBed using ResNet-18, and compare the relative improvement ($\Delta$) over ERM to that of our method (also on DomainBed but using ResNet-50).
>
> **Table R1: Relative improvement ($\Delta$) in accuracy of XDED [1] and DomED over ERM baselines.**
>
> | Method | PACS | OfficeHome | VLCS | TerraInc. | Average |
> | :--- | :---: | :---: | :---: | :---: | :---: |
> | **ResNet-18** | | | | | |
> | ERM (Baseline) | 83.0 | 65.7 | 77.2 | 41.4 | 66.8 |
> | XDED [1] | 83.8 | 65.0 | 74.8 | 42.5 | 66.5 |
> | *$\Delta$ (Gain)* | *+0.8* | *-0.7* | *-2.4* | *+1.1* | *-0.3* |
> | **ResNet-50** | | | | | |
> | ERM (Baseline) | 85.1 | 67.1 | 78.0 | 47.8 | 69.5 |
> | **DomED (Ours)** | **86.9** | **71.2** | **80.1** | **50.2** | **72.1** |
> | *$\Delta$ (Gain)* | *+1.8* | *+4.1* | *+2.1* | *+2.4* | *+2.6* |
>
> The results demonstrate that while XDED struggles to consistently outperform the ERM baseline on some of the benchmarks ($-0.3%$ average improvement), DomED achieves consistent and significant gains across all datasets ($+2.6%$ average improvement). This confirms that distilling from diverse, domain-specific experts is more effective for DG than self-distillation.
>
> **Response to W2: Theoretical Analysis**
>
> We acknowledge that providing a theoretical guarantee for ensemble distillation in the context of DG remains a challenging open problem. However, our design is grounded in the "multi-view" hypothesis for ensemble learning [2], which suggests that ensembles benefit from learning distinct features (views) of the data. In DG, domains naturally provide these distinct views. By training teachers on single, non-overlapping domains, we maximize the diversity of these views. Furthermore, the intuition behind our complementary distillation is to distill the *generalization capability* of the teachers. By evaluating teachers on domains they have not seen, the student learns from their response to distribution shifts, effectively capturing the "disagreement" or uncertainty inherent in OOD inference rather than merely mimicking their training set memorization.
>
> Regarding the decoupled uncertainty loss, we address the optimization conflict identified in previous work [3]. Attempting to simultaneously optimize the mean (accuracy) and the spread (uncertainty) using only the Dirichlet loss often creates conflicting gradients, leading to poor convergence of the mean prediction. Our approach ($\mathcal{L}\_S + \beta \mathcal{L}\_{Dir}$) uses standard cross-entropy to anchor the mean prediction while letting $\mathcal{L}\_{Dir}$ focus solely on shaping the uncertainty. This decoupling allows us to significantly improve calibration without sacrificing the accuracy provided by the standard loss. To further verify this, we conducted an ablation study on the PACS dataset (detailed in Appendix E). As shown in Table R2 below, combining the losses preserves accuracy and significantly improves both calibration (ECE) and rejection capability (PRR) compared to using $\mathcal{L}\_{Dir}$ alone.
>
> **Table R2: Impact of loss components on accuracy and uncertainty metrics evaluated on PACS.**
>
> | Loss Configuration | Accuracy(%) ($\uparrow$) | ECE ($\downarrow$) | NLL ($\downarrow$) | PRR ($\uparrow$) | K.Unc $\uparrow$ |
> | :--- | :---: | :---: | :---: | :---: | :---: |
> | Only $\mathcal{L}\_{Dir}$ | 67.88 | 0.397 | 0.917 | 0.640 | 0.672 |
> | Only $\mathcal{L}\_S$ | 86.89 | 0.080 | 0.542 | 0.771 | 0.719 |
> | **DomED ($\mathcal{L}\_{S} + \beta \mathcal{L}\_{Dir}$)** | **86.94** | **0.044** | **0.473** | **0.787** | **0.723** |

---

> ### Author Response · Authors · 2025-11-27
> **Response to Reviewer XjwQ (2/3)**
>
> **Response to W3: Scalability and Computational Cost**
>
> We appreciate the reviewer raising the issue of scalability. We have conducted a refined cost analysis on the PACS dataset using a single V100 GPU, removing overhead from training monitoring to measure the computational cost more accurately. The results, presented in the table below (and updated in Appendix A.2), demonstrate that DomED is highly efficient.
>
> Notably, the total training time for DomED (2.86 h) is around 50% longer than ERM while achieving significantly better accuracy and calibration. The training of the teacher ensemble is particularly fast (0.78 h) and does not scale linearly with the number of domains. This efficiency stems from two factors: 1) Since each teacher sees only one domain, it is trained with a batch size $1/M$ of the standard ERM batch size, reducing the cost of each forward/backward pass. 2) As discussed in Appendix A.1, effective distillation does not require fully converged teachers because the student model saturates early, which allows us to significantly reduce the number of teacher training steps compared to a standard run.
>
> **Table R3: Computational cost and performance comparison on PACS (Target domain: Sketch).**
>
> | Method | Wall-Time (h) $\downarrow$ | Peak Mem (GB) $\downarrow$ | Acc. (%) $\uparrow$ | ECE $\downarrow$ | NLL $\downarrow$ |
> | :--- | :---: | :---: | :---: | :---: | :---: |
> | ERM | **1.91** | 8.13 | 79.3 | 0.121 | 0.919 |
> | DNA | 3.23 | 8.17 | 79.8 | 0.087 | 0.708 |
> | **DomED** | 2.86 | 8.13 | **81.6** | **0.028** | **0.609** |
> | *DomED (Teachers)* | *0.78* | ***3.00*** | *-* | *-* | *-* |
> | *DomED (Student)* | *2.08* | *8.13* | *-* | *-* | *-* |
>
> Regarding the combination with SWAD, we apply weight averaging only during the distillation phase of the student model. This incurs a small overhead compared to standard SWAD, but achieves higher accuracy by combining the benefits of ensemble knowledge and flat-minima optimization (see Table 2).
>
> **Table R4: Total training time comparison with SWAD on PACS (Target domain: Sketch).**
>
> | Method | Wall-Time (h) $\downarrow$ |
> | :--- | :---: |
> | SWAD | 2.41 |
> | **DomED + SWAD** | 3.27 |
>
> **Response to W4: Modest Accuracy Gains**
>
> We respectfully point out that DomED is built directly upon the simple ERM baseline, rather than being an incremental addition to already sophisticated state-of-the-art methods. We included the combination with SWAD primarily to ensure a fair comparison with recent baselines that rely on weight averaging techniques. Even without SWAD, DomED achieves significant improvements over ERM and remains highly competitive against other strong standalone baselines (see Table 2).
>
> More importantly, we emphasize that the primary contribution of this work extends beyond marginal accuracy gains. We provide the first systematic investigation of ensemble and distillation strategies tailored specifically for Domain Generalization (Table 1). Our analysis reveals that specific data allocation schemes are crucial for effectiveness. Furthermore, we address the critical but largely overlooked challenge of uncertainty quantification in DG. DomED successfully bridges this gap, delivering a method that achieves both strong generalization accuracy and reliable uncertainty estimates, a dual capability that most existing DG methods fail to offer.

---

> ### Author Response · Authors · 2025-11-27
> **Response to Reviewer XjwQ (3/3)**
>
> **Response to W5: Comparison with KD-based Methods (NKD, RISE, BOLD)**
>
> We thank the reviewer for pointing out these relevant works. We have expanded our evaluation to include NKD [4], RISE [5], and BOLD [6] in the revised manuscript (updated Table 2 and Section 4.2). To ensure a rigorous comparison, we restricted all methods to the standard ResNet-50 setting (excluding external auxiliary models like CLIP). As shown in the Table R5 below, DomED achieves the highest average accuracy among these knowledge distillation-based approaches.
>
> **Table R5: Comparison with knowledge distillation methods.**
>
> | Method | PACS | OfficeHome | VLCS | TerraInc. | DomainNet | **Avg.** |
> | :--- | :---: | :---: | :---: | :---: | :---: | :---: |
> | NKD [4] | 83.3% | 71.1% | 77.1% | 37.2% | 42.4% | 62.2% |
> | RISE [5] | 85.0% | 71.5% | 77.6% | 39.0% | 45.2% | 63.7% |
> | BOLD [6] | 85.7% | **72.6%** | 78.7% | 44.3% | **46.9%** | 65.6% |
> | **DomED (Ours)** | **86.9%** | 71.2% | **80.1%** | **50.2%** | 45.7% | **66.8%** |
>
> References:
>
> [1] K. Lee, S. Kim, and S. Kwak, "Cross-domain ensemble distillation for domain generalization," in ECCV, 2022.
>
> [2] Z. Allen-Zhu and Y. Li, "Towards understanding ensemble, knowledge distillation and self-distillation in deep learning," in ICLR, 2023.
>
> [3] M. Ryabinin, A. Malinin, and M. Gales, "Scaling ensemble distribution distillation to many classes with proxy targets," in NeurIPS, 2021.
>
> [4] Y. Wang, H. Li, L. Chau, and A. Kot, "Embracing the dark knowledge: Domain generalization using regularized knowledge distillation," in ACM Multimedia, 2021.
>
> [5] Z. Huang et al., "A sentence speaks a thousand images: Domain generalization through distilling clip with language guidance," in ICCV, 2023.
>
> [6] D. Zhao et al., "Balancing invariant and specific knowledge for domain generalization with online knowledge distillation," in IJCAI, 2025.

---

### Official Review · Reviewer_i5S4 · 2025-10-30

**Soundness:** 3
**Presentation:** 3
**Contribution:** 3
**Rating:** 6
**Confidence:** 4

**Summary:**

The paper proposes Domain-aware Ensemble Distillation (DomED), a framework of efficient ensemble distillation for domain generalization (DG). DomED investigates and optimizes data allocation schemes for training teacher models on different source domains, followed by complementary domain distillation to a student model. The authors also propose a decoupled uncertainty-preserving distillation objective without sacrificing predictive accuracy. Extensive experiments on several datasets demonstrate DomED's competitive performance and substantial improvements in uncertainty quantification over selected baselines.

**Strengths:**

1. The thorough evaluation of data allocation schemes is insightful, covering all possible variations with single model architecture and data of distinct domains.

2. The rigorous ablations and clear analyses validate the importance of teacher diversity, the effect of each loss component, and the choice of hyperparameters.

3. The writing and organization of the paper is commendable, highlighting the improved efficiency and uncertainty quantification while maintaining the accuracy.

**Weaknesses:**

1. The empirical section lacks a direct comparison with a highly pertinent work XDED [1],  and work [2] should at least be mentioned in related work 4.2, both for completeness.

2. The discussion on the scalability with respect to the number of domains is missing. Since the teacher diversity is critical, it remains unknown whether DomED will degrade significantly when the number of teacher decreases.

3. There are a few minor issues can be improved, e.g., tables could be resized denser, and the notation $\tau_m$ in Equation (3) could be revised to $\tau^m$ for consistency.

[1] *Cross-Domain Ensemble Distillation for Domain Generalization.* in ECCV 2022.

[2] *Ensemble Distillation for Out-of-distribution Detection.* in ICPADS 2023.

**Questions:**

1. Have the author compared DomED’s computational cost in terms of training wall-time and memory usage with other single-model DG baselines such as DNA?

2. Is there the risk of negative transfer when some teachers are 'amateur', i.e., trained on domains with very limited data? At what point might teacher diversity become detrimental?

Please respond to both the weaknesses and the questions for rebuttal.

---

> ### Author Response · Authors · 2025-11-27
> **Response to Reviewer i5S4 (1/2)**
>
> We sincerely thank Reviewer i5S4 for the thorough evaluation and for recognizing the rigor of our ablations and the clarity of our presentation. We have incorporated your suggestions to improve the manuscript and address your specific questions below.
>
> **Response to W1: Comparison with XDED and Related Work**
>
> We appreciate the suggestion to discuss XDED [1] and EKD [2]. We have updated Section 4.2 to include these works.
> Methodologically, XDED self-distills an ensemble of the output logits from training data with the same label, whereas DomED leverages an explicit ensemble of domain-specialized teachers to maximize diversity for distillation. For empirical comparison, we note that the absence of publicly available code for XDED makes a direct side-by-side comparison difficult. Nevertheless, we take their reported results on DomainBed using ResNet-18, and compare the relative improvement ($\Delta$) over ERM to that of our method (also on DomainBed but using ResNet-50).
>
>
> **Table R1: Relative improvement ($\Delta$) in accuracy of XDED [1] and DomED over ERM baselines.**
>
> | Method | PACS | OfficeHome | VLCS | TerraInc. | Average |
> | :--- | :---: | :---: | :---: | :---: | :---: |
> | **ResNet-18** | | | | | |
> | ERM (Baseline) | 83.0 | 65.7 | 77.2 | 41.4 | 66.8 |
> | XDED [1] | 83.8 | 65.0 | 74.8 | 42.5 | 66.5 |
> | *$\Delta$ (Gain)* | *+0.8* | *-0.7* | *-2.4* | *+1.1* | *-0.3* |
> | **ResNet-50** | | | | | |
> | ERM (Baseline) | 85.1 | 67.1 | 78.0 | 47.8 | 69.5 |
> | DomED (Ours) | 86.9 | 71.2 | 80.1 | 50.2 | 72.1 |
> | *$\Delta$ (Gain)* | *+1.8* | *+4.1* | *+2.1* | *+2.4* | *+2.6* |
>
> The results demonstrate that while XDED struggles to consistently outperform the ERM baseline on some of the benchmarks ($-0.3%$ average improvement), DomED achieves consistent and significant gains across all datasets ($+2.6%$ average improvement). This confirms that distilling from diverse, domain-specific experts is more effective for DG than self-distillation. Regarding EKD [2], we have clarified in the updated related work that its primary focus is binary OOD detection, whereas DomED targets multi-class generalization and calibration.
>
> **Response to W2: Scalability with Respect to the Number of Domains**
>
> We agree that analyzing scalability is essential to understanding the role of teacher diversity. To quantify the impact of ensemble size, we conducted an ablation study on the PACS dataset. We systematically train models on all possible 2-domain subsets ($M=2$, each resulting in 2 domain-specific teachers) and compare their average performance against the standard 3-domain configuration ($M=3$) as well as an ERM baseline trained on the same 2-domain subsets. The results are summarized below (and included in Appendix B.3):
>
> **Table R2: Impact of ensemble size ($M$) on student performance evaluated on PACS.**
>
> | Metric | DomED ($M=3$) | DomED ($M=2$) | ERM ($M=2$) |
> | :--- | :---: | :---: | :---: |
> | Accuracy $\uparrow$ | 86.9% $\pm$0.2 | 84.6% $\pm$0.3 | 84.3% $\pm$0.2 |
> | ECE $\downarrow$ | 0.044 | 0.050 | 0.097 |
> | NLL $\downarrow$ | 0.473 | 0.504 | 0.649 |
> | PRR $\uparrow$ | 0.787 | 0.701 | 0.649 |
>
> These results lead to two key observations: 1) Despite the reduction in teachers, DomED ($M=2$) still outperforms the ERM baseline trained on the identical data subset. While accuracy is comparable, DomED maintains significantly better calibration (ECE 0.050 vs. 0.097). 2) Reducing the ensemble size from 3 to 2 leads to a decrease in the Prediction Rejection Ratio (0.787 $\to$ 0.701) and a slight increase in ECE. This confirms that the diversity provided by a larger set of domain experts is indeed a primary driver of high-quality uncertainty estimation.
>
> This indicates that while performance benefits from scaling up the number of domains, the method remains effective and robust when domain availability is limited.
>
> **Response to W3: Minor Issues (Notation & Formatting)**
>
> We thank the reviewer for the detailed attention to detail. We have revised the notation $\tau_m$ to $\tau^m$ in Equation (3) for consistency. Additionally, we have compacted the layout of Tables 3, 4, and the appendix tables to improve readability and presentation.

---

> ### Author Response · Authors · 2025-11-27
> **Response to Reviewer i5S4 (2/2)**
>
> **Response to Q1: Computational Cost vs. Single-Model Baselines**
>
> We appreciate the reviewer raising this important point. We have conducted a refined cost analysis on the PACS dataset using a single V100 GPU, removing overhead from training monitoring to measure the computational cost more accurately. The results, presented in the table below (and updated in Appendix A.2), demonstrate that DomED is highly efficient.
>
> Notably, the total training time for DomED (2.86 h) is lower than DNA [3] (3.23 h) while achieving higher accuracy and better calibration (ECE 0.028 vs 0.087). The training of the teacher ensemble is particularly fast (0.78 h) and does not scale linearly with the number of domains. This efficiency stems from two factors: 1) Since each teacher sees only one domain, it is trained with a batch size $1/M$ of the standard ERM batch size, reducing the cost of each forward/backward pass. 2) As discussed in Appendix A.1, effective distillation does not require fully converged teachers because the student model saturates early, which allows us to significantly reduce the number of teacher training steps compared to a standard run.
>
> **Table R3: Computational cost and performance comparison on PACS (Target domain: Sketch).**
>
> | Method | Wall-Time (h) $\downarrow$ | Peak Mem (GB) $\downarrow$ | Best Acc. (%) $\uparrow$ | ECE $\downarrow$ | NLL $\downarrow$ |
> | :--- | :---: | :---: | :---: | :---: | :---: |
> | ERM | **1.91** | 8.13 | 79.3 | 0.121 | 0.919 |
> | DNA | 3.23 | 8.17 | 79.8 | 0.087 | 0.708 |
> | **DomED** | 2.86 | 8.13 | **81.6** | **0.028** | **0.609** |
> | *DomED (Teachers)* | *0.78* | ***3.00*** | *-* | *-* | *-* |
> | *DomED (Student)* | *2.08* | *8.13* | *-* | *-* | *-* |
>
> **Response to Q2: Risk of Negative Transfer from "Amateur" Teachers**
>
> To evaluate robustness against weak teachers trained on limited data, we performed a stress test on PACS (target domain: Sketch) by reducing the training data of specific domains to 50% and 25% (see Appendix C), simulating "amateur" teachers. We compared our standard strategy (equal contribution) against a down-weighted strategy, where the weak teacher's contribution to the soft target was manually suppressed (lowering the weight from $0.8$ to $0.2$).
>
> **Table R4: Robustness of different weighting strategies to data scarcity evaluated on PACS (Target domain: Sketch).**
>
> | Data-Scarce Domain | Data % | Strategy | Student Acc (%) $\uparrow$ | ECE $\downarrow$ |
> | :--- | :---: | :--- | :---: | :---: |
> | **None (Baseline)** | 100% | Standard | 81.6% $\pm$0.2 | 0.028 |
> | **Art Painting** | 50% | Standard | 81.4% $\pm$0.3 | 0.042 |
> | | | Down-weighted | 80.8% $\pm$0.2 | 0.049 |
> | | 25% | Standard | 80.5% $\pm$0.4 | 0.050 |
> | | | Down-weighted | 80.6% $\pm$0.3 | 0.061 |
> | **Photo** | 50% | Standard | 79.8% $\pm$0.3 | 0.054 |
> | | | Down-weighted | 80.2% $\pm$0.2 | 0.064 |
> | | 25% | Standard | 78.0% $\pm$0.5 | 0.041 |
> | | | Down-weighted | 76.5% $\pm$0.4 | 0.069 |
> | **Cartoon** | 50% | Standard | 77.8% $\pm$0.3 | 0.029 |
> | | | Down-weighted | 77.5% $\pm$0.2 | 0.027 |
> | | 25% | Standard | 78.6% $\pm$0.4 | 0.041 |
> | | | Down-weighted | 76.9% $\pm$0.3 | 0.035 |
>
> The results show that DomED is remarkably resilient. Even with 25% data, the performance drop is relatively small. Furthermore, manual down-weighting often degraded accuracy and calibration (higher ECE). This suggests that even "amateur" teachers provide valuable diversity that aids generalization. The ensemble averaging mechanism, combined with the student's hard-label loss anchor, naturally mitigates the noise from weaker teachers without requiring manual intervention.
>
> References:
>
> [1] K. Lee, S. Kim, and S. Kwak, "Cross-domain ensemble distillation for domain generalization," in ECCV, 2022.
>
> [2] D. Wang et al., "Ensemble Distillation for Out-of-distribution Detection," in ICPADS, 2023.
>
> [3] X. Chu et al., "DNA: Domain generalization with diversified neural averaging," in ICML, 2022.

---

### Official Review · Reviewer_PTLs · 2025-10-31

**Soundness:** 3
**Presentation:** 3
**Contribution:** 2
**Rating:** 4
**Confidence:** 4

**Summary:**

This paper proposes a new ensemble distillation method for domain generalization. By combining standard distillation loss and Dirichlet NLL loss, the distilled model can achieve both good accuracy and epistemic uncertainty estimation. The authors provide comprehensive analysis on data allocation strategies for ensemble distillation and compare with multiple baselines on standard benchmarks.

**Strengths:**

1. This paper studies an important problem in ensemble learning for domain generalization. The authors focus on both accuracy and uncertainty estimation.

2. The authors provide comprehensive analysis on the data allocation strategies for ensemble distillation. This is valuable.

3. The authors recognize the problems in Dirichlet-based uncertainty estimation method and propose Decoupled Uncertainty Distillation to address it.

**Weaknesses:**

1. Training different models on different domains is relatively straightforward. The main novelty lies in the complementary distillation and decoupled uncertainty learning.

2. By distilling into a single student model, the method loses the flexibility to adaptively weight ensemble members differently for different test domains. This may create potential challenges for handling diverse distribution shifts since one model cannot fit all.

3. The authors claim the Dirichlet objectives can conflict with accuracy, forcing the model to compromise. But what the authors actually do is very straightforward: just adding the standard distillation loss to enforce accurate mean prediction. But this will make the uncertainty estimation not accurate, right?

**Questions:**

1. You add L_S to keep accuracy. But doesn't this hurt uncertainty quality? How do you verify the uncertainty is still accurate?

2. DiWA requires models trained on all domains and uses 60 checkpoints for weight averaging. Your teachers are trained on single domains only. Do you use the same base models for comparison? If not, how can we know if the performance difference comes from the method or the training setup?

3. Different test domains may require different ensemble weights for each model. How does your single distilled student handle this limitation?

---

> ### Author Response · Authors · 2025-11-27
> **Response to Reviewer PTLs (1/2)**
>
> We sincerely thank Reviewer PTLs for the constructive feedback and for recognizing the value of our comprehensive analysis on data allocation and our proposed complementary distillation approach. We address your specific concerns and questions below.
>
> **Response to W1: Novelty and Significance of the Data Allocation Scheme**
>
> We agree that training models on different domains is conceptually straightforward. However, we employ this "one-teacher-per-domain" strategy as a deliberate design choice to minimize training overhead while maximizing teacher diversity, which is crucial for both OOD generalization and uncertainty estimation.
>
> Our analysis (Appendix B.1) reveals that single-domain teachers exhibit greater specialization. This specialization creates the necessary variance in predictions when evaluating on OOD data. This variance provides the signal that the Dirichlet loss ($\mathcal{L}\_{Dir}$) captures to estimate epistemic uncertainty. As shown in Table 3, reducing this diversity (e.g., by training teachers on multiple domains) degrades the student's calibration performance. Thus, this specific allocation scheme is crucial for the decoupled uncertainty distillation to work effectively. We have revised Section 3.2 to clarify this point.
>
> **Response to W2 & Q3: Flexibility vs. a Single Student Model**
>
> We acknowledge that distilling an ensemble into a single student loses the ability to dynamically re-weight members at test time. However, this trade-off takes into account the standard domain generalization protocol, which typically assumes inference without access to test-domain metadata or batches required for test-time adaptation [1, 2].
>
> Instead of dynamic weighting, our student model "internalizes" the ensemble's consensus during training. By learning from the aggregated soft targets of teachers on domains they have not seen (complementary distillation), the student captures a robust, generalizable representation that performs well on average across unseen shifts. The strong performance on both OOD generalization and uncertainty quantification demonstrates that this internalized robustness is an effective and efficient strategy for standard DG tasks.

---

> ### Author Response · Authors · 2025-11-27
> **Response to Reviewer PTLs (2/2)**
>
> **Response to W3 & Q1: The Accuracy-Uncertainty Conflict and Verification**
>
> The reviewer raises a valid concern regarding whether adding the standard cross-entropy loss ($\mathcal{L}\_S$) compromises uncertainty estimation. Interestingly, our empirical results suggest that $\mathcal{L}\_S$ is essential for stabilizing the optimization. Minimizing the Dirichlet NLL ($\mathcal{L}\_{Dir}$) alone attempts to simultaneously optimize the mean prediction (accuracy) and the concentration (uncertainty). In the challenging DG setting, this often leads to poor convergence of the mean prediction [3]. By introducing $\mathcal{L}\_S$ as an anchor for the mean prediction, we allow $\mathcal{L}\_{Dir}$ to focus primarily on shaping the distribution's spread.
>
> To verify this, we conducted an ablation study on the PACS dataset (detailed in Appendix E). As shown in Table R1 below, combining the losses significantly improves both calibration (ECE) and rejection capability (PRR) compared to using $\mathcal{L}\_{Dir}$ alone.
>
> **Table R1: Impact of loss components on accuracy and uncertainty metrics evaluated on PACS.**
>
> | Loss Configuration | Accuracy(%) ($\uparrow$) | ECE ($\downarrow$) | NLL ($\downarrow$) | PRR ($\uparrow$) | K.Unc $\uparrow$ |
> | :--- | :---: | :---: | :---: | :---: | :---: |
> | Only $\mathcal{L}\_{Dir}$ | 67.88 | 0.397 | 0.917 | 0.640 | 0.672 |
> | Only $\mathcal{L}\_S$ | 86.89 | 0.080 | 0.542 | 0.771 | 0.719 |
> | **DomED ($\mathcal{L}\_S + \mathcal{L}\_{Dir}$)** | **86.94** | **0.044** | **0.473** | **0.787** | **0.723** |
>
> These results confirm that $\mathcal{L}\_S$ does not degrade uncertainty; rather, it provides an auxiliary learning signal that allows the uncertainty objective to function correctly.
>
> **Response to Q2: Fairness of Comparison with DiWA**
>
> We used the standard pre-trained ResNet-50 backbone (same as DiWA) for all experiments to ensure a fair comparison. The performance difference stems from the training methodology rather than the base model. DiWA relies on weight averaging over 60 checkpoints trained on all domains, which is computationally intensive. In contrast, DomED uses $M$ domain-specific teachers, with a total teacher training cost even smaller than training a single standard model (see Appendix A for computational cost analysis).
>
> To provide a fair comparison regarding the benefits of weight averaging, we included results for DomED+SWAD in Table 2. This combination allows us to compare our distillation approach against DiWA's averaging approach on equal footing. The results show that DomED is highly competitive when combined with SWAD, thereby highlighting the efficiency of our ensemble distillation framework.
>
> References:
>
> [1] H. Yao et al., "Improving domain generalization with domain relations," in ICLR, 2024.
>
> [2] H. Yan and Y. Guo, "Context-aware self-adaptation for domain generalization," arXiv:2504.03064, 2025.
>
> [3] M. Ryabinin, A. Malinin, and M. Gales, "Scaling ensemble distribution distillation to many classes with proxy targets," in NeurIPS, 2021.

---

### Comment · Area_Chair_jmUw · 2025-11-20
**To AI Review**

Dear authors,

I would like to share an important reminder regarding the review process. Recently, we have noticed that some reviewers may be using AI tools to help generate their reviews. This can lead to low-quality or inaccurate feedback, which is unfair to authors who deserve careful and thoughtful evaluations.

To help maintain fairness, I kindly ask for your assistance: If you believe a review you received was partly or fully generated by AI, and you have some evidence (for example: unusual writing style, clear factual mistakes, AI-detector results, repeated generic sentences, etc.), please feel free to contact me directly.

I will review any evidence you provide and, if appropriate, adjust the weight of the reviewer’s evaluation so that it does not negatively affect your submission. Thank you for helping us keep the review process fair and responsible. Your understanding and cooperation are greatly appreciated.

Best regards,

AC

---

### Comment · Area_Chair_jmUw · 2025-11-27

Dear Reviewers and Authors,

As we are approaching the rebuttal deadline, I would like to share a gentle reminder with everyone.

For authors:
If you have not yet submitted your rebuttal, please make sure to do so as soon as possible. Submitting very close to the deadline may reduce the chance for reviewers to read and respond in time, which could affect the discussion phase.

For reviewers:
If a rebuttal has already been submitted for your assigned papers, I encourage you to take a moment to read it and, where appropriate, provide a brief response or update your evaluation. Of course, this is not meant to pressure anyone into changing scores, it is simply to ensure that all reviews remain well-informed before final decisions.

Thank you all for your time and effort in keeping the review process smooth and constructive.

Warm regards,
AC

---

### Author Response · Authors · 2025-12-01
**Summary of Rebuttal and Revisions (1/3)**

Dear AC,

We understand the challenges posed by the recent changes regarding the review process and sincerely appreciate the time and effort you are dedicating to evaluating our work. We are encouraged by the reviewers' recognition of our systematic analysis of ensemble and distillation strategies for domain generalization (DG), the focus on both accuracy and uncertainty, the rigorous ablation studies, and the clarity of the presentation.

To assist in your decision-making, we provide a summary of how we have addressed the reviewers' feedback. In response to the comments from all four reviewers, we have significantly updated our manuscript, adding approximately 6 pages of new content (mainly in Section 3 and Appendices A–E) containing extensive additional experiments and analyses.

Below, we summarize our responses to the shared concerns regarding novelty, computational cost, and the validity of our decoupled objective.

### 1. Novelty and Comparison with Existing Methods (XDED)

Reviewers **i5S4** and **XJwQ** requested a comparison with XDED [1] and questioned the novelty of our ensemble strategy.

While both involve distillation, they use fundamentally different mechanisms. XDED is a self-distillation method that distills an ensemble of the output logits (of the same model) from training data with the same label, *ignoring the domain labels*. In contrast, DomED constructs an explicit ensemble of domain-specialized teachers by using the domain labels. This difference is crucial for both generalization and calibration, since a single model (XDED) cannot capture the "disagreement" between diverse experts.

For empirical comparison, we note that the absence of publicly available code for XDED makes a direct side-by-side comparison difficult. Nevertheless, we take their reported results on DomainBed using ResNet-18, and compare the relative improvement ($\Delta$) over ERM to that of our method (also on DomainBed but using ResNet-50).

**Table R1: Relative improvement ($\Delta$) in accuracy of XDED [1] and DomED over ERM baselines.**

| Method            | PACS     | OfficeHome | VLCS     | TerraInc. | Average  |
|:----------------- |:--------:|:----------:|:--------:|:---------:|:--------:|
| **ResNet-18**     |          |            |          |           |          |
| ERM (Baseline)    | 83.0     | 65.7       | 77.2     | 41.4      | 66.8     |
| XDED [1]          | 83.8     | 65.0       | 74.8     | 42.5      | 66.5     |
| *$\Delta$ (Gain)* | *+0.8*   | *-0.7*     | *-2.4*   | *+1.1*    | *-0.3*   |
| **ResNet-50**     |          |            |          |           |          |
| ERM (Baseline)    | 85.1     | 67.1       | 78.0     | 47.8      | 69.5     |
| **DomED (Ours)**  | **86.9** | **71.2**   | **80.1** | **50.2**  | **72.1** |
| *$\Delta$ (Gain)* | *+1.8*   | *+4.1*     | *+2.1*   | *+2.4*    | *+2.6*   |

The results demonstrate that while XDED struggles to consistently outperform the ERM baseline on some of the benchmarks ($-0.3%$ average improvement), DomED achieves consistent and significant gains across all datasets ($+2.6%$ average improvement). This confirms that distilling from diverse, domain-specific experts is more effective for DG than self-distillation.

---

### Author Response · Authors · 2025-12-01
**Summary of Rebuttal and Revisions (2/3)**

### 2. Computational Efficiency and Scalability

Reviewers **i5S4** and **XJwQ** raised concerns regarding the training cost of multiple teachers and scalability.

We have conducted a refined cost analysis on the PACS dataset using a single V100 GPU, removing overhead from training monitoring to measure the computational cost more accurately. The results, presented in the table below (and updated in Appendix A.2), demonstrate that DomED is highly efficient.

**Table R2: Computational cost and performance comparison on PACS (Target domain: Sketch).**

| Method             | Wall-Time (h) $\downarrow$ | Peak Mem (GB) $\downarrow$ | Best Acc. (%) $\uparrow$ | ECE $\downarrow$ | NLL $\downarrow$ |
|:------------------ |:--------------------------:|:--------------------------:|:------------------------:|:----------------:|:----------------:|
| ERM                | **1.91**                   | 8.13                       | 79.3                     | 0.121            | 0.919            |
| DNA                | 3.23                       | 8.17                       | 79.8                     | 0.087            | 0.708            |
| **DomED**          | 2.86                       | 8.13                       | **81.6**                 | **0.028**        | **0.609**        |
| *DomED (Teachers)* | *0.78*                     | ***3.00***                 | *-*                      | *-*              | *-*              |
| *DomED (Student)*  | *2.08*                     | *8.13*                     | *-*                      | *-*              | *-*              |

Notably, the total training time for DomED (2.86 h) is lower than DNA [2] (3.23 h) while achieving higher accuracy and better calibration (ECE 0.028 vs 0.087). The training of the teacher ensemble is particularly fast (0.78 h) and does not scale linearly with the number of domains. This efficiency stems from two factors: 1) Since each teacher sees only one domain, it is trained with a batch size 1/M of the standard ERM batch size, reducing the cost of each forward/backward pass. 2) As discussed in Appendix A.1, effective distillation does not require fully converged teachers because the student model saturates early, which allows us to significantly reduce the number of teacher training steps compared to a standard run.

---

### Author Response · Authors · 2025-12-01
**Summary of Rebuttal and Revisions (3/3)**

### 3. Validity of the Decoupled Uncertainty Objective

Reviewer **PTLs** asked if adding the standard cross-entropy loss ($\mathcal{L}_S$) compromises uncertainty estimation, and Reviewer **XJwQ** asked for theoretical justification.

We acknowledge that providing a theoretical guarantee for ensemble distillation in the context of DG remains a challenging open problem. However, our design is grounded in the "multi-view" hypothesis for ensemble learning [3], which suggests that ensembles benefit from learning distinct features (views) of the data. In DG, domains naturally provide these distinct views. By training teachers on single, non-overlapping domains, we maximize the diversity of these views. Furthermore, the intuition behind our complementary distillation is to distill the *generalization capability* of the teachers. By evaluating teachers on domains they have not seen, the student learns from their response to distribution shifts, effectively capturing the "disagreement" or uncertainty inherent in OOD inference rather than merely mimicking their training set memorization.

Regarding the decoupled uncertainty loss, we address the optimization conflict identified in previous work [4]. Attempting to simultaneously optimize the mean (accuracy) and the spread (uncertainty) using only the Dirichlet loss often creates conflicting gradients, leading to poor convergence of the mean prediction. Our approach ($\mathcal{L}\_S + \beta \mathcal{L}\_{Dir}$) uses standard cross-entropy $\mathcal{L}\_S$ to anchor the mean prediction while letting $\mathcal{L}\_{Dir}$ focus solely on shaping the uncertainty. This decoupling allows us to significantly improve calibration without sacrificing the accuracy provided by the standard loss. To further verify this, we conducted an ablation study on the PACS dataset and present the results in the table below (detailed in Appendix E).

**Table R3: Impact of loss components on accuracy and uncertainty metrics evaluated on PACS.**

| Loss Configuration                              | Accuracy(%) ($\uparrow$) | ECE ($\downarrow$) | NLL ($\downarrow$) | PRR ($\uparrow$) | K.Unc $\uparrow$ |
|:----------------------------------------------- |:------------------------:|:------------------:|:------------------:|:----------------:|:----------------:|
| Only $\mathcal{L}_{Dir}$                        | 67.88                    | 0.397              | 0.917              | 0.640            | 0.672            |
| Only $\mathcal{L}_S$                            | 86.89                    | 0.080              | 0.542              | 0.771            | 0.719            |
| **DomED ($\mathcal{L}\_S + \mathcal{L}\_{Dir}$)** | **86.94**                | **0.044**          | **0.473**          | **0.787**        | **0.723**        |

As shown in the table, combining the losses preserves accuracy and significantly improves both calibration (ECE) and rejection capability (PRR) compared to using $\mathcal{L}_{Dir}$ alone. These results confirm that $\mathcal{L}_S$ does not degrade uncertainty; rather, it provides an auxiliary learning signal that allows the uncertainty objective to function correctly.

### Conclusion

We believe the revised manuscript presents a novel, efficient, and robust framework that simultaneously addresses generalization accuracy and uncertainty quantification, a dual capability often missing in existing DG literature. We hope these additional results and clarifications assist you in your final assessment.



**References:**

[1] K. Lee, S. Kim, and S. Kwak, "Cross-domain ensemble distillation for domain generalization," in ECCV, 2022.

[2] X. Chu et al., "DNA: Domain generalization with diversified neural averaging," in ICML, 2022.

[3] Z. Allen-Zhu and Y. Li, "Towards understanding ensemble, knowledge distillation and self-distillation in deep learning," in ICLR, 2023.

[4] M. Ryabinin, A. Malinin, and M. Gales, "Scaling ensemble distribution distillation to many classes with proxy targets," in NeurIPS, 2021.



Sincerely,

Authors

---

### Meta-Review · Area_Chair_da5f · 2026-01-05

**Summary:**

The submission proposes a novel domain generalisation method based on ensemble distillation, with a particular focus on improved calibration. The reviewers were generally positive about the presentation of the work and that the submission investigates the understudied problem of calibration in DG. However, there were some concerns raised several times: the method has limited novelty over existing ensemble distillation approaches; the key contribution (the loss function) is not motivated/justified by any theoretical or empirical analysis; and the experimental comparison is missing many important baselines.

**Reviewer Concerns:**

The rebuttal successfully addresses some of the minor concerns/questions from the reviewers, but the response to the three key concerns above was not sufficient. There are a number of other ensemble distillation methods that should be compared with when targeting DG accuracy. I would also expect to see how combing the most closely related methods with a standard calibration approach, such as Platt scaling, performs on the calibration evaluation.

**Reviewer Scores:**

I do not think the reviewers would have changed their scores.

---

### Decision · Program_Chairs · 2026-01-26

Reject